

# Seasonal dynamics and disturbance of phytoplankton biomass in the wake of Tahiti as observed by Biogeochemical-Argo floats

Raphaëlle Sauzède[1], Elodie Martinez[1], Orens Pasqueron de Fommervault[2], Antoine Poteau[3], Alexandre Mignot[3], Christophe Maes[4], Hervé Claustre[3], Julia Uitz[3], Keitapu Maamaatuaiahutapu[5], Martine Rodier[1], Catherine Schmechtig[3] and Victoire Laurent[6]

[1]Ecosystèmes Insulaires Océaniens (EIO, UMR-241), IRD, Ifremer, UPF and ILM, Tahiti, French Polynesia
[2]Departamento de Oceanografìa Fisica, Centro de Investigacion Cientìfica y de Educacion Superior de Ensenada, Carretera Ensenada-Tijuana 3918, Zona Playitas, Ensenada, BC 22860, Mexico
[3]Sorbonne Universités, UPMC Univ Paris 06, CNRS-INSU, Observatoire Océanologique de Villefranche, Laboratoire d'Océanographie de Villefranche, 181 Chemin du Lazaret, 06230 Villefranche-Sur-Mer, France
[4]Laboratoire d'Océanographie Physique et Spatiale (LOPS), IUEM, Univ. Brest, Ifremer, CNRS, IRD, F-29280, Brest, France
[5]Laboratoire de Géosciences du Pacifique Sud, Université de la Polynésie française, Tahiti, French Polynesia
[6]Direction Inter Régionale de Polynésie Française, Météo France, BP 6005 98702 Tahiti, French Polynesia

*Correspondence to*: Raphaëlle Sauzède (raphaelle.sauzede@ird.fr)

**Abstract.** The South Pacific Subtropical Gyre (SPSG) is a vast and remote area where large uncertainties on variability in phytoplankton biomass and production remain due to the lack of biogeochemical *in situ* observations. In such oligotrophic environments, ecosystems are predominantly controlled by nutrients depletion in surface waters. However, this oligotrophic character can be disturbed in the vicinity of islands where enhancement of biological activity is known to occur (i.e. the island mass effect, IME). This study mainly focuses on *in situ* observations showing that an IME can be evidenced leeward of Tahiti (17.7°S - 149.5°W), French Polynesia. Concomitant physical and biogeochemical observations collected with two Biogeochemical-Argo (BGC-Argo) profiling floats from April 2015 to November 2016 are used to investigate the dynamics of phytoplankton biomass. The first float has a transit of more than 1000 km westward of Tahiti (open ocean conditions) while the second one remained in the Tahitian wake (around 45 km from the island coasts). In the oligotrophic central SPSG, the wintertime increase in upper layer chlorophyll *a* concentration is likely due to photoacclimation process. Vertical observations show a light-driven deepening of the deep chlorophyll maximum (DCM) from winter to summer, consistently with previous descriptions. At the opposite, within the Tahitian wake, the DCM temporary widens during late spring in association with a biological enhancement in the upper layer. Combining *in situ* measurements with meteorological data along the Tahiti coasts, Hybrid Coordinate Ocean Model outputs and satellite-derived products (i.e., horizontal currents and associated fronts), the physical mechanisms involved in the disturbance of phytoplankton seasonal cycle in the Tahitian wake have been investigated. This disturbance results from the concomitant occurrence of strong precipitations and a zone of weak currents leeward Tahiti. We conjecture that the land drainage induces a significant supply of nitrate in the ocean upper layer (down to ~100 m) while a zone of weak currents in the southwestern zone behind Tahiti forms an accumulation zone, hence allowing phytoplankton growth up to 20 km away from the coastlines. Moreover, bio-optical measurements suggest that the composition of



phytoplankton community could differ in the Tahitian wake *vs*. the open ocean area. Finally, in addition to extending information to the water column, only BGC-Argo floats could provide biogeochemical measurements in the SPSG region when clouds prevent the use of remote sensing.

## 1 Introduction

5        Phytoplankton is an essential component in marine biogeochemical cycles because its spatiotemporal distribution drives the trophic structure of marine ecosystems (Iverson, 1990) and constrains the variability and the production of the world ocean's fisheries (Chassot et al., 2010). Oligotrophic environments are commonly defined as regions of the global ocean where surface waters are nutrient depleted, inducing a lack of new production (Hamner and Hauri, 1981), and where surface chlorophyll *a* concentration (Chl), a widely used proxy of phytoplankton biomass, is lower than 0.1 mg m$^{-3}$ (Antoine et al.,

1996). Although oligotrophic regions account for more than 50% of the global ocean, few studies have described the seasonal variability of the vertical distribution of phytoplankton biomass in such environments (Letelier et al., 2004; Mignot et al., 2014).

        The South Pacific Subtropical Gyre (SPSG) is the most oligotrophic ocean (Dandonneau et al., 2006; Morel et al., 2010) and the largest oceanic desert of the planet (Claustre and Maritorena, 2003). The SPSG is also one of the least known

and studied oceanic regions because this area is vast and remote, leading to scarce biogeochemical *in situ* observations (Sauzède et al., 2015) and large uncertainties on phytoplankton biomass variability. This gyre is also distinguishable by its extremely weak sources of nutrients from deep layers (Raimbault et al., 2008), as well as from atmospheric flux (Claustre et al., 2008; Mahowald et al., 2005; Wagener et al., 2008). In these surface nutrient depleted waters, the biological enrichment reported near many islands, the so-called "Island Mass Effect" (IME; Doty and Oguri, 1956), locally enhances to the

productivity and potential fisheries (Gove et al., 2016). By supporting a greater abundance of fish and reef-building organisms in comparison with more oligotrophic waters, the IME-induced nearshore biological enhancement is also critical for coral reef ecosystems development and sustainability (Williams et al., 2015). The IME may result from many processes such as the vertical transport of nutrient-rich water masses (e.g. coastal upwelling, eddy induced mixing and internal waves; e.g. Heywood et al., 1990; Palacios, 2002; Signorini et al., 1999) or island-related inputs (e.g. submarine groundwater discharge, rivers

outflow; e.g. Dandonneau and Charpy, 1985; Perissinotto et al., 2000). A recent Pacific basin-scale study showed that IME is a near-ubiquitous feature among island- and atoll-reef ecosystems and across broad gradients of oceanic conditions (Gove et al., 2016). Unfortunately, this study is limited to the 20°S - 30°N and 140°E - 150°W geographical zone (mainly westward of the dateline), missing the eastern part of the SPSG and associated oligotrophic conditions. Moreover, most studies dealing with the signature of IME on phytoplankton biomass are based on satellite-derived products analyses (e.g. Palacios, 2002; Martinez

and Maamaatuaiahutapu, 2004; Andrade et al., 2014), hence missing the main part of the euphotic layer where phytoplankton photosynthesis takes place (Morel and Berthon, 1989). However, the nearshore enhancement of phytoplankton biomass occurs within the whole euphotic layer (Gove et al., 2016). Thus, evaluating the impact of IME on the vertical distribution of



phytoplankton biomass using adequate *in situ* measurements through the water column remains crucial.

The integration of miniaturized biogeochemical sensors on autonomous profiling floats since the early 2000s now offers a new and promising way of observing hydrological and key biogeochemical properties at appropriate spatial and temporal scales required to develop mechanistic understanding of key biogeochemical processes. The so-called Biogeochemical-Argo (BGC-Argo) floats (Johnson and Claustre, 2016) have been successful in addressing such processes in the open ocean with unprecedented temporal and vertical resolutions (e.g. Boss and Behrenfeld, 2010; Green et al., 2014; Mignot et al., 2014, 2016; Grenier et al., 2015; Sauzède et al., 2016; Zhang et al., 2016; Chacko, 2017; and references therein). More specifically for oligotrophic environments, Mignot et al. (2014, hereafter referred as M2014) revealed the potential of such autonomous platforms to investigate the seasonal dynamics of the vertical distribution of phytoplankton biomass. Although this study reports the only description of the seasonal dynamics of the vertical distribution of phytoplankton biomass in the SPSG, it has however some limitations. Indeed, M2014 only covered one year of measurements (over 2009) and focused on the eastern South Pacific ultra-oligotrophic environment, missing for instance the central oligotrophic part of the SPSG. Moreover, due to limitations in the float equipment, some fundamental variables were not measured *in situ*, such as the Photosynthetically Available Radiation (PAR, that was derived from downward irradiance measured at 490 nm) and nitrate concentration, both the main drivers for phytoplankton growth (Cullen, 2015).

In 2015 and 2016, two BGC-Argo floats were deployed leeward of Tahiti island (17.7° S, 149.5° W), French Polynesia (Figure 1). They experienced two distinct pathways. One float drifted in the open ocean more than 1000 km westward away from Tahiti coasts to the center of the SPSG (referred as FOpenO) while the second float remained within 45 km off Tahiti, under the influence of the Tahitian wake (referred as FTWake). These two pathways allowed the observation of phytoplankton biomass dynamics over a broad range of scales from seasonal to shorter time scales as well as in the open ocean and in an island wake. The present study has three main objectives: (1) to investigate the seasonal dynamics in phytoplankton biomass in the open ocean conditions in the center of the SPSG; (2) to compare the open ocean dynamics and variability with the ones observed in the Tahitian wake; (3) to characterize and identify the key mechanisms impacting on the vertical distribution of phytoplankton biomass and resulting in the signature of the Tahitian IME.

## 2 Data and Methods

### 2.1 Biogeochemical-Argo measurements

In this study, we use two BGC-Argo floats, FOpenO and FTWake, also identified from their World Meteorological Organization (WMO) numbers as 6901687 and 6901659, respectively (http://www.argo.ucsd.edu, http://argo.jcommops.org). The two floats are equipped with a Seabird standard conductivity-temperature-depth (CTD) and completed with additional sensor packages. The Satlantic OCR radiometer measures the downwelling irradiance at three wavelengths (380 nm, 410 nm and 490 nm) as well as the PAR; the WET Labs ECO Puck Triplet is composed of a chlorophyll *a* fluorometer, a Colored Dissolved Organic Matter (CDOM, see acronym definitions in Table 2) fluorometer, and a sensor that measures the



backscattering coefficient at 700 nm. FTWake is additionally equipped with an oxygen Aanderaa optode (model 4330) that measures the oxygen concentration and a Satlantic Submersible Ultraviolet Nitrate Analyzer (SUNA) sensor that measures the nitrate concentration. The mission parameters and equipment relative to each float used in this study are summarized in Table 1 whereas their respective trajectory is presented in Figure 1.

The fluorescence of chlorophyll *a* can be considered as a proxy for chlorophyll *a* concentration (Cullen, 1982, 2015). Nevertheless the relationship between chlorophyll *a* concentration (Chl) and fluorescence is highly variable and depends on phytoplankton community composition and physiology as well as light and nutrient conditions (e.g. Cleveland and Perry, 1987; Roesler and Barnard, 2013; Sackmann et al., 2008). In this study, each profile of chlorophyll *a* fluorescence is converted into Chl following several considerations. First, the BGC-Argo recommended protocol described by Schmechtig et al. (2014; 1, 2

and 8 quality control flags are used) is applied to each profile. The instrumental dark signal and negative spikes are removed from fluorescence profiles. Moreover, the so-called daytime non-photochemical quenching (NPQ) process that results in a decrease of phytoplankton fluorescence per unit of Chl and occurs at high irradiance (Cullen and Lewis, 1995), is corrected according to the method developed by Xing et al. (2012), using a depth threshold of 90 % of the mixed layer depth (see details in Schmechtig et al., 2014). Then, the recommended global bias correction factor of 2 is applied to each calibrated fluorescence

profile (Roesler et al., 2017). Finally, profiles are also corrected from the fluorescence originated from non-algal matter as recently suggested by Xing et al. (2017). *In situ* Chl determinations from High Performance Liquid Chromatography (HPLC) analyses were performed during the FTWake deployment. The Chl values derived from the fluorescence profile measured at the FTWake float deployment are strongly correlated with those obtained from the HPLC analyses ($R^2$ = 0.92 and slope of 1.27 using a linear regression, Figure S1a). This supports the robustness of the procedure applied for calibrating the BGC-

Argo fluorescence measurements.

       The backscattering sensors implemented on BGC-Argo floats measure the angular scattering coefficient at 124° relative to the direction of light propagation and at a wavelength of 700 nm. This measurement is then transformed into the particulate backscattering coefficient at 700 nm, $b_{bp}(700)$ (hereafter $b_{bp}$) following Schmechtig et al. (2016). As described by Sauzède et al. (2016), a quality control procedure is then applied to each $b_{bp}$ profile by removing: (1) high frequency spikes by

using a 7-point running median filter and (2) $b_{bp}$ values outside of the sensor operation range (> 0.03 m$^{-1}$). Then, $b_{bp}$ values can be interpreted in terms of Particulate Organic Carbon stock (POC; Loisel et al., 2001, 2002, Stramski et al., 1999, 2008). Specifically, Stramski et al. (2008) established a relationship between POC and $b_{bp}(555)$ in the SPSG based on measurements performed as part of the BIOSOPE cruise (Claustre et al., 2008). Here, to estimate POC from $b_{bp}(700)$ measurements acquired by the two BGC-Argo floats, $b_{bp}(700)$ is converted into $b_{bp}(555)$ using the commonly used power law model of the particulate

backscattering spectral dependency (e.g. Loisel et al., 2006; Reynolds et al., 2001; Stramska et al., 2003):

$$b_{bp}(\lambda) = b_{bp}(\lambda_0) \cdot \left(\frac{\lambda}{\lambda_0}\right)^{-\gamma} \tag{1}$$

with γ=2.5. This value is derived from *in situ* $b_{bp}$ measurements in the SPSG, at different wavelengths from the BIOSOPE dataset (during austral summer 2004). This γ value is in agreement with the study of Loisel et al. (2006) that associated low





chlorophyll waters of subtropical gyres with high γ values (between 2 and 3) resulting from a dominance of small-sized particles. Finally, from $b_{bp}(555)$ estimates and using the POC *vs.* $b_{bp}(555)$ slope developed from the BIOSOPE dataset excluding upwelling data (Stramski et al., 2008), $b_{bp}$ is converted into POC for FOpenO and FTWake.

The dissolved oxygen concentration ($O_2$) is derived from the Aanderaa optode measurements following the procedure described by Thierry et al. (2016). Based on the pre-deployment in-air measurements from FTWake, a factor of 1.07 and no offset are applied to each $O_2$ profile (Bittig and Körtzinger, 2015).

The SUNA sensor allows quantifying the nitrate concentration ($NO_3^-$) from light absorption at ultraviolet wavelengths (Johnson et al., 2010, 2013; Johnson and Coletti, 2002). As recommended by Johnson et al. (2016), $NO_3^-$ is calculated using the algorithm developed by Sakamoto et al. (2009). All data used in this study are also pressure-corrected as suggested first by Pasqueron de Fommervault et al. (2015) and confirmed by laboratory measurements (Johnson et al., 2016, 2017a, 2017b). A post-deployment calibration is finally performed to correct data from instrumental drift. Each nitrate profile is then adjusted using a reference value at depth (i.e. 900 - 950 m), estimated with an accuracy of 0.65 µmol kg$^{-1}$ from the recently developed CANYON method (for CArbonate system and Nutrients concentration from hYdrological properties and Oxygen using a Neural-network; Sauzède et al., 2017). The first $NO_3^-$ profile measured from FTWake is compared with *in situ* $NO_3^-$ estimates (from SEAL-AA3 continuous Flow Analyzer) and shows very good consistency ($R^2 = 0.97$ and slope of 1.02 using a linear regression, Figure S1b).

The CTD data are quality controlled following the standard Argo protocol (Wong et al., 2015). The mixed layer depth (MLD) is estimated according to the difference in potential density (σ) between 10 m and the base of the mixed layer using a threshold of 0.125 kg m$^{-3}$ (a reference for studies regarding subtropical gyres; e.g. Levitus, 1982; Ohno et al., 2004; Suga et al., 2004; Nicholson et al., 2015; Toyoda et al., 2015).

The cell residence time in the mixed layer during a convection event is proportional to the MLD at the first order (Mignot et al., 2016, see their equation B4). In the extremely deep ML in the North Atlantic (800 – 1000 m), phytoplankton cells complete one revolution per day in the ML during convective mixing (D'Asaro, 2008). Then, since MLD in the SPSG are much shallower than in the North Atlantic, we can reasonably assume that the cells make a revolution in less than a day in the ML, suggesting that within a given day, all phytoplankton cells experience all light conditions in the ML. Consequently, such as described by Mignot et al. (2017) the average radiant energy that phytoplankton cells received in a day is obtained by averaging the vertical profile of daily PAR over the depth of the mixed layer ($PAR_{ML}$).

Chl and $b_{bp}$ are calculated from the surface to the first penetration depth ($Z_{pd}$, a reference depth for ocean color remote sensing studies) and referred as $Chl_{surf}$ and $b_{bp\_surf}$. $Z_{pd}$ is here defined as $1/K_d(PAR)$ (Gordon and McCluney, 1975) with $K_d(PAR)$ being the diffuse attenuation coefficient for PAR and derived from the following equation:

$$PAR(z) = PAR(0^-)e^{-K_d(PAR)*z} , \qquad (2)$$

where $PAR(0^-)$ is the PAR just below the sea surface. Finally, the euphotic depth, $Z_{eu}$, is calculated as the depth where the PAR is reduced to 1% of its surface value.





### 2.2 HYCOM modelled ocean currents

Ocean current dataset from daily archive of the 1/12° reanalysis of the Hybrid Coordinate Ocean Model (HYCOM, http://hycom.org; Chassignet et al., 2009) using the Navy Coupled Ocean Data Assimilation (NCODA) system is used to provide an overview of the currents in the study area and along the floats trajectories. The NCODA system (Cummings, 2005;

Cummings and Smedstad, 2014) assimilates available satellite altimeter observations, satellite and *in situ* Sea Surface Temperature (SST) as well as available *in situ* vertical temperature and salinity profiles (from XBTs, Argo floats and moored buoys). The GLBu0.08/expt_91.1 experiment is hereafter used. Ocean currents (i.e., speed as well as the zonal and meridian components) are extracted from the surface down to 300 m over the central SPSG, bounded by 170°W- 140°W and 10°S- 30°S, and from April 2015 to November 2016.

### 2.3 Altimetry-derived products

To analyze fronts, transport barriers and horizontal stirring induced by surface currents, we investigate the Finite Size Lyapunov Exponents (FSLEs) that are passive Lagrangian diagnostics derived from ocean surface geostrophic currents (e.g. d'Ovidio et al., 2004; Mancho et al., 2008). FSLEs are currently provided by AVISO+ (Archiving Validation and Interpretation of Satellite Oceanographic Data, www.aviso.altimetry.fr) with a spatial resolution of 1/25° and a temporal resolution of 4 days.

These data are extracted over the same region and period as described in Section 2.2 for the HYCOM dataset.

### 2.4 Cumulative precipitations data

The daily cumulative precipitation data are provided by the French Meteorological Institute in French Polynesia. Data were acquired from 4 meteorological stations located along the south-eastern coast of Tahiti (i.e., stations Afaahiti 3, Teahupoo 1, Teahupoo 2 and Vairao 2, locations are detailed in Section 3.3.2).

## 3 Results and Discussion

The very different drifts experienced by each float allow addressing a broad range of spatial and temporal scales with respect of phytoplankton biomass dynamics (Figure 1). Results are thus presented and analyzed along three distinct parts. First, FOpenO provides indications on the seasonal dynamics of phytoplankton biomass and associated hydrological properties in the central SPSG over more than 18 months. ~3 months of measurements during spring and early summer from FTWake allow

investigating the specific phytoplankton biomass dynamics in the Tahitian wake. Thus, in the second part, a comparison between both dynamics in the open ocean and in the Tahitian wake has been performed. In the final part, the key mechanisms impacting on the vertical distribution of phytoplankton biomass in the Tahitian wake have been identified and characterized.





### 3.1 Dynamics of phytoplankton biomass in the central SPSG

The seasonal dynamics of the Chl vertical distribution and associated density in the open ocean are evaluated from FOpenO (Figure 2). The Deep Chlorophyll Maximum (DCM, calculated as the depth where Chl is maximum) is permanently established and follows isolumes (red and black lines in Figure 2a, respectively). Applying a moving average (±5 observations)

filtering to minimize the effect of mesoscale variability, the depths of the DCM and of the specific isolume of 1 mol photons $m^{-2}$ $d^{-1}$ are significantly correlated (r = 0.89 and p-value < 2.2 $10^{-16}$ using a Pearson's test). The DCM reaches its deepest position of ~150 m simultaneously to its highest Chl values of ~ 0.35 mg $m^{-3}$ during austral spring/early summer (i.e. from September to January 2015 and 2016; Figure 2a and Table 3 for seasonal filtered values).

The wintertime shallowing of the DCM from June to September, associated with an increase in Chl in the 0 – 50 m

upper layers, is concomitant with the deepest ML (Figure 2 and Table 3). From the surface to the first penetration depth ($Z_{pd}$ in Figure 3a, blue line), the wintertime $Chl_{surf}$ is maximum while $b_{bp\_surf}$ is low and $b_{bp\_surf}^{*}$, the ratio of $b_{bp\_surf}$ to $Chl_{surf}$, is minimum (Figures 3b to e, blue line). The $b_{bp}$ signal is dominated by sub-micronic (0.1-10 µm size range) particles and phytoplankton cells (Behrenfeld et al., 2005; Huot et al., 2007, 2008; Loisel et al., 2007; Morel and Ahn, 1991). It has been shown to be a good estimate of phytoplankton carbon and is used to track changes in phytoplankton biomass (Behrenfeld et

al., 2005; Graff et al., 2015; Martinez-Vicente et al., 2013). Thus, $b_{bp\_surf}^{*}$ allows tracking changes in phytoplankton physiology resulting from photoacclimation (Behrenfeld et al., 2005; Behrenfeld and Boss, 2003, 2006; M2014), i.e., the increase (decrease) in phytoplankton intracellular chlorophyll *a* content with the decrease (increase) in average light (Letelier et al., 1993; Morel et al., 2010; Winn et al., 1995). It is here worth mentioning the tight relationships between the seasonal variations in $b_{bp}^{*}$ (Figure 3e) and the average irradiance within the ML ($PAR_{ML}$, Figure 3f). During winter, when $b_{bp\_surf}^{*}$ is minimum,

$PAR_{ML}$ is strongly reduced (up to a factor of about 3 with respect to summer conditions; Figure 3f). During this period, the weakening in surface irradiance intensity is concomitant to enhanced mixing, resulting in a severe reduction in light availability. Therefore, the observed winter increase in $Chl_{surf}$ is the consequence of photoacclimation rather than of an increase in phytoplankton carbon biomass.

The only comparable existing study in the SPSG performed by M2014 is based on a one-year (2009) time-series in

the eastern ultra-oligotrophic part of the SPSG and on derived-PAR estimates. Here, we confirm these previous results over a longer and different time period (i.e. April 2015 to November 2016), in the central oligotrophic SPSG, and using *in situ* PAR measurements. We consistently find a seasonal displacement of the DCM well related to isolumes and a wintertime Chl increase in the upper layer likely resulting from photoacclimation.

### 3.2 Comparison between the central SPSG and the Tahitian wake

Surface measurements in the open ocean from FOpenO and in the Tahitian wake from FTWake have been superimposed in Figure 3. Maximum $b_{bp\_surf}$ measured from FTWake is twice higher than for FOpenO (~ 0.06 $10^{-2}$ $m^{-1}$ *vs.* ~ 0.03 $10^{-2}$ $m^{-1}$; Figure 3d). This suggests a significant increase in POC, most likely in phytoplankton biomass, in the Tahitian





wake compared with open ocean conditions in spring and summer 2015/2016. This enhancement of phytoplankton biomass leeward of Tahiti is not visible from remote sensing due to a high cloud coverage during the rainy season all over the region.

FOpenO and FTWake show similar light conditions ($PAR_{ML}$; Figure3f). Therefore, discrepancies between $b_{bp\_surf}{}^*$ in FOpenO and FTWake (Figure 3e) cannot be related to photoacclimation, but likely reflect differences in phytoplankton

community composition (or in the nature of the particle assemblage) in the island wake as compared to the open ocean. Phytoplankton community composition in the SPSG is usually dominated by pico- and nano-phytoplankton (e.g. Ras et al., 2008). Cetinić et al. (2015) showed that $b_{bp}{}^*$ can be used as a proxy for community composition with high values associated with pico- and nano- to low values associated with diatom-dominated communities. Although the SPSG trophic conditions are very different from that investigated by Cetinić et al. (2015) in the North Atlantic Subpolar Gyre, and that $b_{bp}{}^*$ can be also

affected by a change in the composition (size and nature) of the particle assemblage, the change in $b_{bp\_surf}{}^*$ by a factor of 2 between the average FTWake and FOpenO (Figure 3d) suggests a phytoplankton community with a significant contribution of microphytoplankton in the Tahitian wake. Such an increase could be attributed to diatoms, that grow relatively fast and have high affinities for nitrates (Edwards et al., 2012; Glibert et al., 2016; Lomas and Glibert, 2000), and could be in agreement with an increased input of nutrients in the Tahitian wake.

To investigate if the differences in the biogeochemical dynamics between the open ocean and the Tahitian wake conditions observed at the surface can be extended to the water column, monthly mean vertical profiles of Chl and $b_{bp}$ are derived from the two floats (Figure 4). In spring and summer (i.e. from September to March), the DCM depth recorded by FOpenO is almost constant and reaches its deepest location while Chl and $b_{bp}$ intensify until reaching a maximum in summer (Figure 4, red lines on upper and lower left panels), reflecting a maximum of biomass. In the Tahitian wake, an inversion of

this DCM seasonal dynamics can be evidenced. Indeed, the DCM deepens and widens towards the surface layer instead of sharpening, with low Chl and $b_{bp}$ values in December and January (Figure 4, purple and red lines on the upper and lower right panel, respectively). The $b_{bp}$ vertical distribution from FOpenO displays a deep $b_{bp}$ maximum, similarly as the DCM, whereas $b_{bp}$ profiles within the Tahitian wake exhibit a vertical distribution more homogeneous. Thus, the disturbance described above in the surface seasonal dynamics of phytoplankton biomass in the Tahitian wake, is extended to depth leading to an unexpected

vertical distribution of phytoplankton biomass in December and January. In the next section, this disturbance is further investigated through the key biogeochemical parameters observed from the FTWake float, such as the $NO_3{}^-$ concentration.

### 3.3 Evidence of an island mass effect

### 3.3.1 Effect on phytoplankton biomass

The seasonal disturbance in the Tahitian wake phytoplankton biomass is further investigated through the analysis of

the time-series collected by FTWake (Figure 5). Three distinct time periods can be defined according to the Chl and $b_{bp}$ variability in the upper lit surface layer (unfiltered time series from Figure 3c and f are presented in Figures 5a and b).

The first-time period is associated with the expected seasonal transition between the end of austral winter and the



beginning of spring, as observed in open ocean conditions. In October, high $Chl_{surf}$ and low $b_{bp\_surf}$ values (Figure 5a) are associated with deep MLD (Figure 5h) and low $NO_3^-$ concentrations in the upper layer (i.e. < 1 µmol $kg^{-1}$ in the 0 – 100 m layer, Figure 5f). These high $Chl_{surf}$ values may be associated with photoacclimation resulting from reduced $PAR_{ML}$ (~10 mol photons $m^{-2}$ $d^{-1}$; Figure 5b). In November, the MLD shallows and the $Chl_{surf}$ decreases. This $Chl_{surf}$ weakening is only disrupted in early

and late November in association with a $b_{bp\_surf}$ increase (Figure 5a), suggesting an increase in phytoplankton biomass. This point will be discussed in the next section.

The second-time period exhibits a concomitant increase of $Chl_{surf}$ and $b_{bp\_surf}$ (Figure 5a), suggesting a phytoplankton biomass enhancement. This pattern is consistently associated with an enhancement of $NO_3^-$ in the 0 -100 m upper layer characterized by high values up to ~1.4 µmol $kg^{-1}$ (Figures 5d and f). The $NO_3^-$ concentrations are higher at the surface and

decrease with depth down to ~100 m (Figure 5f). The POC, derived from $b_{bp}$ values, is also enhanced in the surface layer by a factor of ~4.5 (mean values over the surface to the $Z_{pd}$ layer; ~30 mg $m^{-3}$ from FOpenO to ~140 mg $m^{-3}$ from FTWake; data not shown) and by a factor of 1.5 when integrated over the whole euphotic layer (from ~6000 mg $m^{-2}$ to ~9000 mg $m^{-2}$; data not shown). The vertical distribution of Chl is also disturbed as the DCM shallows, stops following isolumes (Figures 5c and e), and widens diapycnally (Figure 5g). In fact, the DCM seems to be associated with specific isopycnes all over the FTWake

lifetime, except during the second-time period where it expands over a wide range of isopycnal surfaces (Figure 5g). Over this period, the DCM does not follow isolumes and, thus, does not appear light-driven as usually observed in the SPSG waters. Using a moving average (± 5 observations), the depths of the DCM and of the isolume of 1 mol photons $m^{-2}$ $d^{-1}$ are not significantly correlated (r = 0.32, p-value = 0.0024 using a Pearson's test).

During the third-time period, the $Chl_{surf}$ shows a rapid decrease concomitantly with $b_{bp\_surf}$ and $NO_3^-$ in the upper layer

(Figures 5a and f). The DCM quickly deepens (from ~100 m down to ~150 m; Figure 5c), accompanied by a deepening of nitrate isocontours (Figure 5d), returning to more standard conditions comparable with open ocean conditions in the SPSG.

### 3.3.2 Underlying physical drivers

The disturbance of the classical seasonal cycle when the float is the closest to Tahiti (~20 km off the coastlines) suggests an influence of the island. Enhanced phytoplankton biomass in the island wakes can be the result of specific processes

uplifting nutrient-rich deep waters (e.g. coastal upwelling, eddy induced mixing; e.g. Heywood et al., 1990; Palacios, 2002; Signorini et al., 1999), terrestrial nutrient discharge due to land drainage (e.g. Dandonneau and Charpy, 1985; Perissinotto et al., 2000), or atmospheric dust deposition (e.g. Martino et al., 2014). During the second period, when the $NO_3^-$ observed from FTWake accumulates at the surface and decreases with depth (Figure 5f), the isopycnal depths remain stable (Figure 5h), suggesting that the nitrate enhancement does not come from a vertical uplift. Using satellite-derived products of sea level

anomaly, we ensure that the FTWake float is not under the influence of any eddy structures. The MLD remains shallow during this period (~ 40 m; Figure 5h) and cannot reach the nitracline (~120 m, calculated from isoline of 1 µmol $kg^{-1}$; Figure 5d) to inject deep nutrients into the upper layer.

Next we evaluate the scenario of strong precipitations over Tahiti and associated soil leaching. Such run off of



sediments and other terrigenous nutrient-rich material may have fertilized the offshore waters by horizontal advection. The FTWake temperature/salinity (T-S) diagrams from the surface down to ~200 m, show a relative decrease in sea surface salinity over periods 2 and 3 (Figures 6a and 6b, respectively), likely reflecting the signature of rain events. Consistently, daily cumulative measurements of rain along the southeastern coasts of Tahiti confirm intense precipitations over the FTWake drift

time period (Figure 7). During the period 1, precipitations are also strong (up to 300 mm at the end of November 2015) but the concomitant observed $b_{bp\_surf}$ increases remain weak (<0.05m$^{-1}$; Figure 5a) as compared with the period 2 (up to 0.14 m$^{-1}$). There is no associated $NO_3^-$ enrichment in the upper layer (Figure 5f), likely due to a more distant location of the float from Tahiti (Figure 7a). Nutrients may have been taken up by biota (i.e. supply < consumption), thus, preventing the float to capture the impact of land drainage.

Over the second period, strong precipitations up to 120 mm occur while the float reaches its closest position from Tahiti coastlines (Figure 7a). While these conditions are consistent with the assumption of nitrate supply into the upper layer resulting from land drainage, the float is still about 20 km off the island. The possibility that nutrients can have accumulated and been advected far enough offshore to be recorded by FTWake remains to be explained. Oceanic currents from the HYCOM model reanalysis reveal currents above 10 cm s$^{-1}$ and up to 60 cm s$^{-1}$ during periods 1 and 3 (Figure 8a) along the FTWake

track. Conversely, during period 2, the float crossed a zone of weak current with velocities lower than 10 cm s$^{-1}$, while the SEC mainly flows south-westward in the upper layers, from Tahiti toward the open ocean (Figures 8b and 8c). This supports the possibility that nitrates could have accumulated leeward of Tahiti and then been advected toward the float position. Another process could also explain the broad vertical extension down to ~100 m of the nitrate enhancement (~1.4 μmol kg$^{-1}$). Intense (horizontal and/or vertical) currents flushing out of the passes of the lagoon could advect nutrient-enhanced waters toward the

open ocean. In Moorea (the closest island westward off Tahiti, Figure 1), the current flushing out of the pass has an average of 50 cm s$^{-1}$ and can even reach 200 cm s$^{-1}$ (Lenhardt, 1991). Such phenomena cannot be described by the present HYCOM reanalysis which does not take into account the specific lagoon dynamics and exchanges with the open ocean waters. A more dedicated work based on lagoon modeling is required to explore further this possibility.

During the period 3, strong precipitations still occur over Tahiti but no phytoplankton biomass signature seems visible

as the float drifts away from the island (Figure 5a). There is a clear change in the ocean dynamics between the period 2 to 3. The float flows back to a strong current branch of the SEC, with southwestward velocities stronger than 30 cm s$^{-1}$ down to 300 m and up to 60 cm s$^{-1}$ in its core (Figure 8a). Comparing T-S diagrams between period 2 and 3, a change in water mass can be evidenced below the DCM depth around 150 m (Figure 6a *vs*. 6b). This is also visible in the vertical distribution of $O_2$ with the incursion of a more oxygenated water mass likely coming from the south during period 3 in the 200 - 300 m layer (Figure

9). HYCOM currents at 300 m show a southwestward orientation close to Tahiti when considering the date of transition from period 2 to 3 (Figure 8). However, it is necessary to consider ocean currents with a sufficient time lag to allow the water mass to reach Tahiti before the transition time period. When considering a location south of Tahiti, for instance point A in Figure 10, the time to reach the float position (point B) is about 24 days. Regarding 24 days before the incursion of more oxygenated waters (28/12/2015), the HYCOM currents at the 04/12/2015 date suggest a southern origin of the oxygenated water mass.



Finally, we also investigated satellite derived FSLEs which provide information about fronts, transports barriers and horizontal stirring by surface currents. Figures 11a and 11b show that the float crosses a front coming from the east during the transition between the period 2 to 3 (dark grey filament east and along the crosses, respectively). This shows the rapid and intense evolution of the water mass, evidenced from $O_2$ and TS diagrams, and the fact that the float quickly leaves the accumulation

area of nutrients to return back to a nutrient-depleted environment (Figures 5b, c and f), typical of SPSG oligotrophic conditions. It should be added that it remains always difficult to interpret observations of different nature, i.e., from a Lagrangian vs. Eulerian point of view, and that further work is required to replace the float observations in the full dynamical context since different dynamical regimes and fronts are barriers to transport (Mendoza and Mancho, 2010).

## 5 Summary and conclusion

Two BGC-Argo profiling floats deployed in the central SPSG provided an unprecedented source of information in this scarcely sampled and remote area. These observations were used to describe the seasonal dynamics of phytoplankton biomass into two distinct dynamical environments, namely in the SPSG open ocean and in the Tahitian wake. The open ocean observations confirm the only previous study describing the seasonal vertical dynamics of phytoplankton biomass in the eastern ultra-oligotrophic SPSG from one BGC-Argo float deployed in 2009 (M2014). Our results allowed to go further in the

description of the vertical distribution of phytoplankton biomass by considering another oligotrophic environment (i.e. the central SPSG) and over a longer time series of 2 years (2015 and 2016). We confirm that the wintertime Chl increase in the upper layer is likely due to photoacclimation and that he DCM vertical position is light-driven in this open ocean environment, using *in situ* PAR measurements acquired from FOpenO float (*vs.* PAR derived from downward irradiance as in M2014).

In the close wake of Tahiti, we show that the seasonal dynamics of phytoplankton biomass is disturbed during several

weeks in late spring 2015. This disturbance is associated with an increase in phytoplankton biomass, POC and $NO_3^-$ concentrations within the 100 m upper layer. During this biological enhancement, the DCM shallows and widens toward the surface so that the vertical position of the DCM is not light-driven. The $b_{bp}^*$ optics-based index suggests a shift in phytoplankton community composition from a pico- and nano-dominated community in the open ocean toward a micro-dominated community within the Tahitian wake. It could also suggest changes in the composition (size and nature) of the particle

assemblage rather than in the phytoplankton community structure only. Combining *in situ* float observations with meteorological data on Tahiti, ocean modeling and satellite derived products, the physical mechanisms involved in the disturbance of phytoplankton seasonal variability have been investigated. A concomitant occurrence of land drainage induced by strong precipitations over the island with an area of weak southwestward currents would have maintained the supply of nutrients leeward of Tahiti allowing the float to record the induced biogeochemical enhancement. A more dedicated work

based on lagoon modeling would help to explore further the nitrate supply up to 100 m depth since HYCOM does not take into account the specific lagoon dynamics and exchanges with the open ocean waters.





For the first time, the signature of an IME has been evidenced in a low nutrient low chlorophyll environment based on *in situ* observations collected from BGC-Argo floats. The BGC-Argo float in the Tahitian wake has allowed to describe a phenomenon that cannot be captured by satellites due to the high cloud coverage in the SPSG during the rainy season. The missing satellite data prevented the delineation of the IME geographical extension but thanks to *in situ* float measurements,

the biological enhancement has been evidenced up to 20 km off Tahiti. Moreover, BGC-Argo floats provide the subsurface Chl dynamics, associated with some physical, biogeochemical and bio-optical parameters that cannot be viewed from remote sensing. Mechanisms involved in the IME are complex and vary from an island to another. The BGC-Argo floats, used conjointly with model outputs and satellite-derived products, have been shown to be adequate tools to study IME thanks to their multi-parameters acquisition and high vertical and temporal sampling resolution. In the SPSG, climate change is likely

to impact open ocean food webs and oceanic fisheries (Bell et al., 2011). It is therefore important to document changes in biogeochemical processes using long observational time series and BGC-Argo profiling floats appear to be powerful tools to follow important ecological and biogeochemical changes, especially in an under-sampled area by classical ship-based oceanographic cruises such as the SPSG.

The present study demonstrates that nearshore phytoplankton enhancement can be significant and impact the vertical

distribution of phytoplankton biomass and probably its associated community composition in oligotrophic areas such as the SPSG. In oligotrophic environments, where ecosystems are predominantly based on nutrient-depleted waters, the increase in phytoplankton biomass driven by the IME is critical for coral reef ecosystems (Williams et al., 2015) and fishery productivity but also contributes to carbon vertical flow (Heywood et al., 1996). Finally, such biological enhancement has to be more carefully described and studied in order to be integrated into global biogeochemical models, and hopefully to better predict the

evolution of nearshore marine ecosystems.

**Acknowledgements**

First, we would like to thank the Government of French Polynesia and the French State for providing funding support to the THOT project (Contrat de projets Etat-Pays, convention n°8690/MSR/REC, and arrêté de subvention n° HC/2860/DIE/BPT), including the post doctorate fellowship of R. Sauzède. This work was also supported by the French

national program LEFE/INSU and the French Ministry of the Overseas. FOpenO was provided by the remOcean project (funded by the European Research Council, grant agreement 246777). We also thank the French BGC-Argo program (funded by CNES-TOSCA, LEFE Cyber and LEFE-GMMC) and the Coriolis program for providing standard Argo floats to be equipped with additional bio-optical and biogeochemical sensors. The BGC-Argo data were collected and made freely available by the International Argo Program and the national programs that contribute to it. (http://www.argo.ucsd.edu,

http://argo.jcommops.org/). This Argo Program is part of the Global Ocean Observing System. We would also like to thank Henry Bittig expertise for the calibration (slope and offset determination) of FTWake $O_2$ profiles. AVISO+ (Archiving Validation and Interpretation of Satellite Oceanographic Data, www.aviso.altimetry.fr) freely provided Finite Size Lyapunov

Exponent data used in this study. The 1/12 deg global HYCOM+NCODA Ocean Reanalysis was funded by the U.S. Navy and the Modeling and Simulation Coordination Office. Computer time was made available by the DoD High Performance Computing Modernization Program. The output is publicly available at http://hycom.org. Finally, data from the BIOSOPE transect of particulate backscattering coefficient at different wavelengths were taken from the BIOSOPE database available on the website: http://www.obs-vlfr.fr/proof/php/x_datalist.php?xxop=biosope&xxcamp=biosope; and access have been provided by C. Schmechtig.

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





**Table 1. Summary of the mission and equipment relative to each BGC-Argo float used in this study.**

| Float | WMO # | Length of the mission | UTC date of deployment (dd/mm/YYYY) | Position of deployment (lon/lat) | UTC date of last profile (dd/mm/YYYY) | Temporal resolution of data acquisition | Equipment |
|---|---|---|---|---|---|---|---|
| FOpenO | 6901687 | 19 months | 03/04/2015 | -149.7/-18.24 | 25/11/2016 | -daily from 03/04/2015 to 16/07/2015 -5 days from 16/07/2015 to 25/11/2016 | -CTD -Radiometry -WET Labs ECO Puck Triplet |
| FTWake | 6901659 | ~3 months | 15/10/2015 | -149.1/-18.20 | 10/01/2016 | daily | -CTD -Radiometry -WET Labs ECO Puck Triplet -Nitrate sensor (SUNA) -Oxygen sensor (Aanderaa optode) |



**Table 2. Symbols of variables used in this study with their definitions and units.**

| Symbol | Definition | Units |
|---|---|---|
| PAR | Photosynthetically Available Radiation | mol photons $m^{-2}$ day$^{-1}$ |
| Chl | Chlorophyll *a* concentration | mg $m^{-3}$ |
| $b_{bp}$ | Particulate backscattering coefficient at 700 nm | $m^{-1}$ |
| $b_{bp}^*$ | Ratio of $b_{bp}$ to Chl | $m^2$ $mg^{-1}$ |
| $O_2$ | Oxygen concentration | µmol $kg^{-1}$ |
| $NO_3^-$ | Nitrate concentration | µmol $kg^{-1}$ |
| MLD | Mixed Layer Depth | m |
| $\sigma$ | Potential Density computed from Temperature and Salinity | kg $m^{-3}$ |
| $PAR_{ML}$ | Average value of PAR in the mixed layer depth | mol photons $m^{-2}$ day$^{-1}$ |
| $Chl_{surf}$ | Mean of Chl in the first penetration layer (from surface to $Z_{pd}$ depth) | mg $m^{-3}$ |
| $b_{bp\_surf}$ | Mean of $b_{bp}$ in the first penetration layer (from surface to $Z_{pd}$ depth) | $m^{-1}$ |
| $b_{bp\_surf}^*$ | Mean of $b_{bp}^*$ in the first penetration layer (from surface to $Z_{pd}$ depth) | $m^2$ $mg^{-1}$ |
| $Z_{pd}$ | First penetration depth | m |
| $K_d(PAR)$ | diffuse attenuation coefficient | $m^{-1}$ |
| $PAR(0^-)$ | PAR just below the surafce | mol photons $m^{-2}$ day$^{-1}$ |
| $Z_{eu}$ | Euphotic depth | m |
| FSLE | Finite Size Lyapunov Exponents | d$^{-1}$ |
| T | Conservative Temperature | °C |
| S | Absolute Salinity | g $kg^{-1}$ |



**Table 3. Number of profiles, mean and range values in brackets of mean-filtered (±5 observations) DCM depth, DCM value, MLD and Chl$_{Surf}$ for each float averaged over the 4 seasons. Summer, fall, winter and spring are defined as January to March, April to June, July to September and October to December time periods, respectively.**

| Parameter | Float | Seasons | | | |
| --- | --- | --- | --- | --- | --- |
| | | Winter | Spring | Summer | Fall |
| $N_{obs}$ | FOpenO | 49 | 30 | 18 | 109 |
| | FTWake | | 78 | 10 | |
| DCM depth | FOpenO | 119 | 138 | 133 | 116 |
| (min – max) | | (107 – 144) | (133 – 150) | (130 – 136) | (99 – 130) |
| | FTWake | | 129 | 145 | |
| | | | (114 – 147) | (137 – 150) | |
| DCM value | FOpenO | 0.27 | 0.30 | 0.32 | 0.25 |
| (min – max) | | (0.25 – 0.30) | (0.27 – 0.33) | (0.29 – 0.35) | (0.21 – 0.32) |
| | FTWake | | 0.30 | 0.24 | |
| | | | (0.20 – 0.39) | (0.20 – 0.26) | |
| MLD | FOpenO | 76 | 49 | 40 | 63 |
| (min – max) | | (34 – 96) | (29 – 93) | (36 – 44) | (32 – 87) |
| | FTWake | | 50 | 35 | |
| | | | (24 – 115) | (33 – 38) | |
| Chl$_{Surf}$ | FOpenO | 0.03 | 0.01 | 0.01 | 0.04 |
| (min – max) | | (0.01 – 0.05) | (0.01 – 0.03) | (0.01 – 0.01) | (0.01 – 0.06) |
| | FTWake | | 0.02 | 0.02 | |
| | | | (0.01 – 0.08) | (0.01 – 0.02) | |



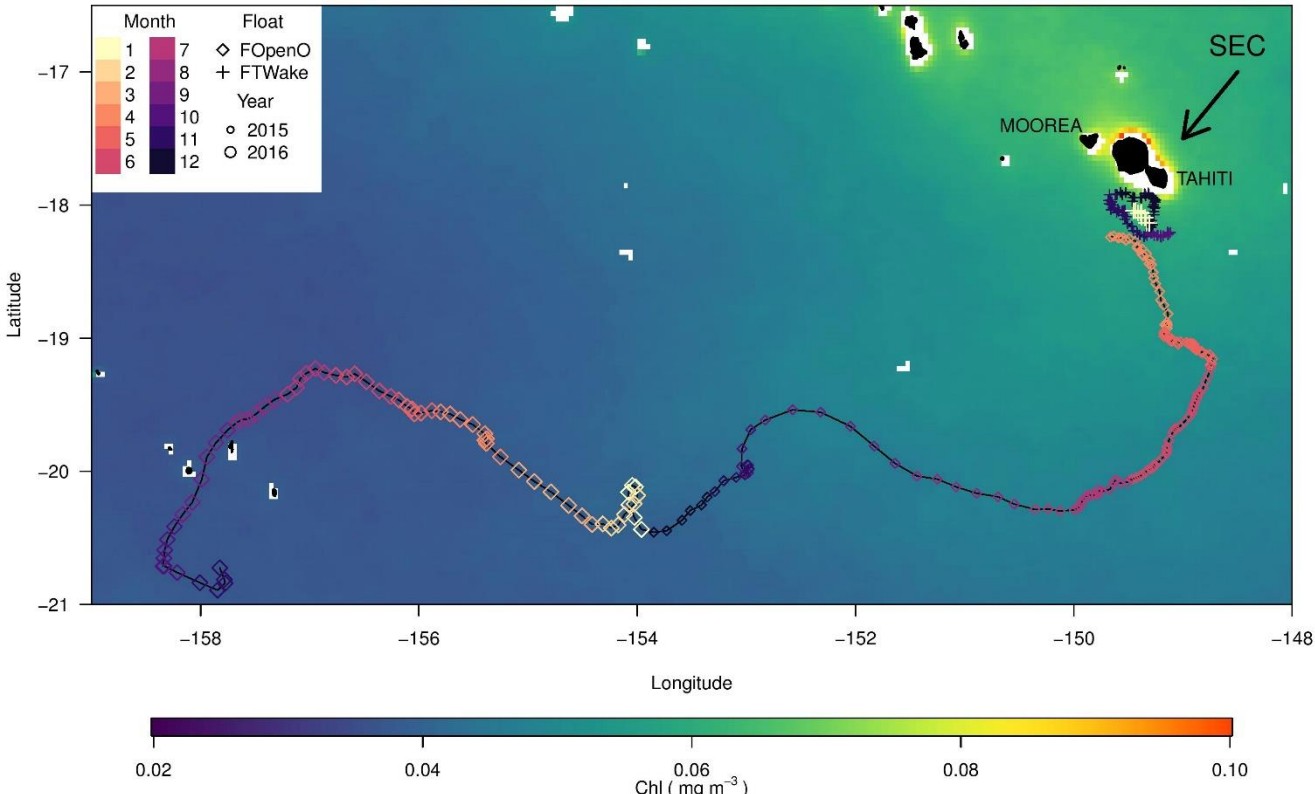

**Figure 1: The trajectories of the two BGC-Argo floats used in this study (differentiated by symbols) are plotted as a function of month (color of the symbols) and year (size of the symbols). The color in background represents the climatological surface satellite Chl (mg m$^{-3}$) over the MODIS Aqua mission (i.e. from 2002 to 2016). The satellite imagery is masked when bathymetry is shallower than 500 m to remove nearshore pixels that could be biased by land (in white). Islands are indicated in black. The Moorea and Tahiti islands are indicated and the mean direction of the South Equatorial Current (SEC) is indicated by the arrow.**



**Figure 2: FOpenO vertical distribution along time of (a) Chl (mg.m⁻³) with the 0.1, 1 and 10 isolumes (mol photons m⁻² d⁻¹, dotted black lines) and (b) density (kg m⁻³) with isopycnes (interval=0.5 kg m⁻³, dotted black lines). In each panel, the white and red lines show the MLD and the location of the DCM, respectively.**





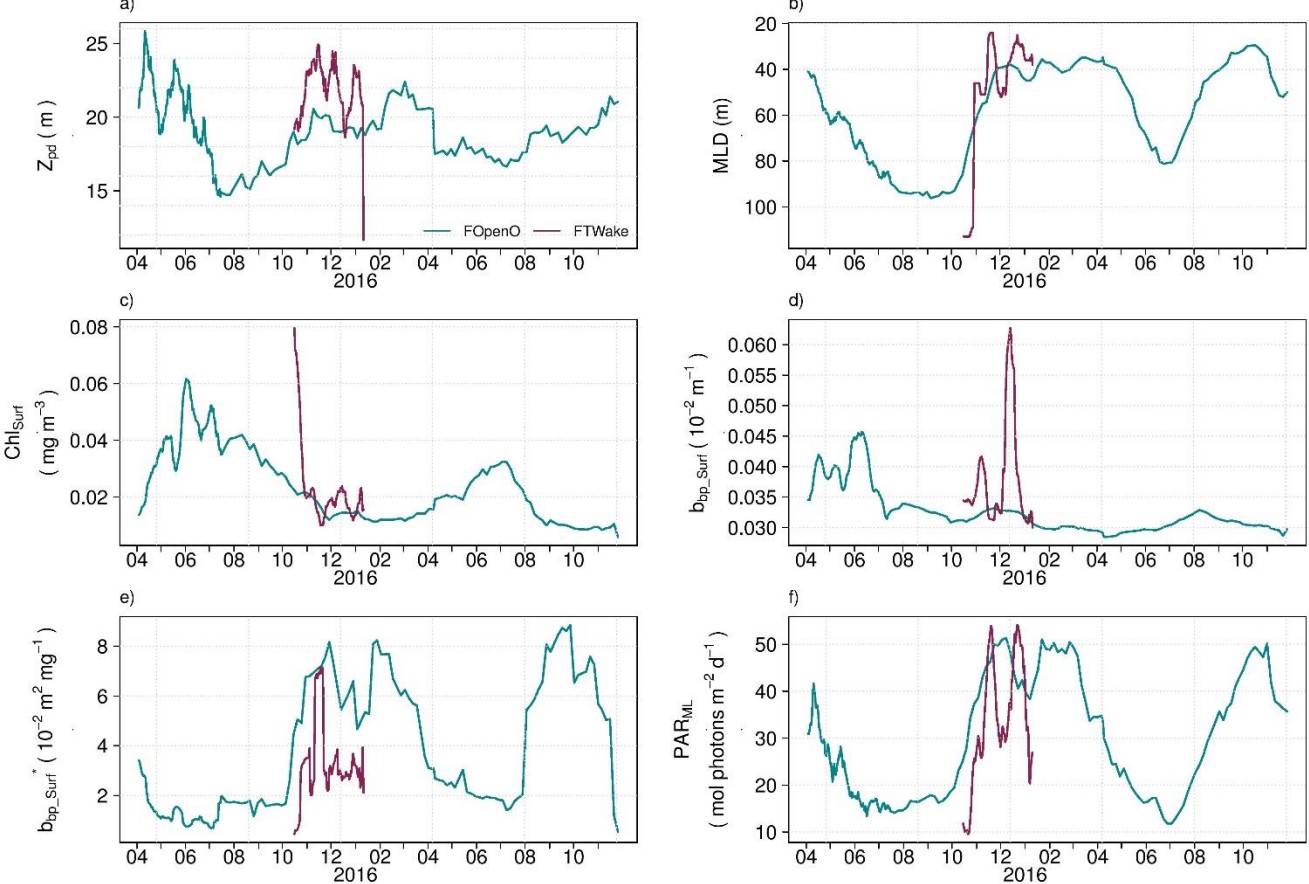

**Figure 3**: **Time series of the mean-filtered (±5 observations) (a) first penetration depth ($Z_{pd}$, m), (b) mixed layer depth (MLD, m), (c) surface Chl ($Chl_{surf}$, mg m$^{-3}$), (d) surface $b_{bp}$ ($b_{bp\_surf}$, m$^{-1}$), (e) surface ratio of $b_{bp}$ to Chl ($b_{bp\_surf}^*$, m$^2$ mg$^{-1}$), and (f) average PAR in the mixed layer depth ($PAR_{ML}$, mol photons m$^{-2}$ d$^{-1}$). FOpenO observations are shown as blue lines and FTWake as purple lines.**





**Figure 4**: **Vertical distribution of monthly average (from top to bottom): Chl (mg m⁻³) and b$_{bp}$ (m⁻¹). The left column represents the FOpenO observations and the right column represents the FTWake observations. The color code represents months over 2015 to 2016.**





**Figure 5**: **Biogeochemical parameters measured from FTWake. Time series of (a) Chl$_{surf}$ (left axis) and b$_{bp\_Surf}$ (right axis) and (b) PAR$_{ML}$. Vertical distribution along time of: (c) Chl (mg.m$^{-3}$) with the 0.1, 1 and 10 isolumes (mol photons m$^{-2}$ d$^{-1}$, black lines), (d) nitrate concentration (µmol L$^{-1}$) with isonitrates (µmol L$^{-1}$, black lines with interval=1 µmol L$^{-1}$). (e) and (f) panels represent the 0 - 150 m zoom of (c) and (d) panels. (g) Time series of Chl (mg.m$^{-3}$) as a function of density (kg.m$^{-3}$). (h) Vertical distribution of density (kg m$^{-3}$) along time with isopycnes (kg m$^{-3}$, black lines with interval=0.5 kg m$^{-3}$). The white and red lines represent the MLD and the location of the DCM depth, respectively in panels (c), (d) and (h). The black vertical solid lines in each panel indicate the three specific time periods as defined in Section 3.3.1.**



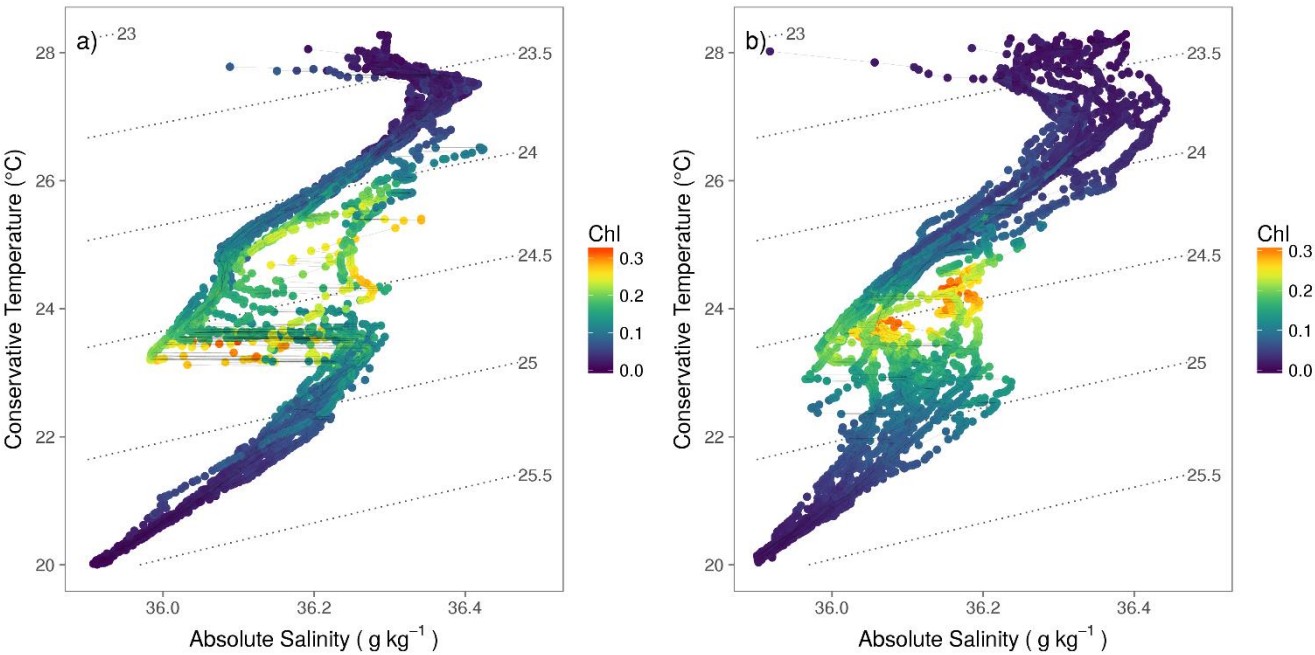

**Figure 6: T-S diagrams issued from the FTWake during (a) period 2 and (b) period 3 as defined in the text (see Section 3.3.1) and Figure 5. Black dotted lines represent isopycnal surfaces (interval=0.5 kg m⁻³). Chl (mg m⁻³) associated with the T-S measurements is shown in color.**



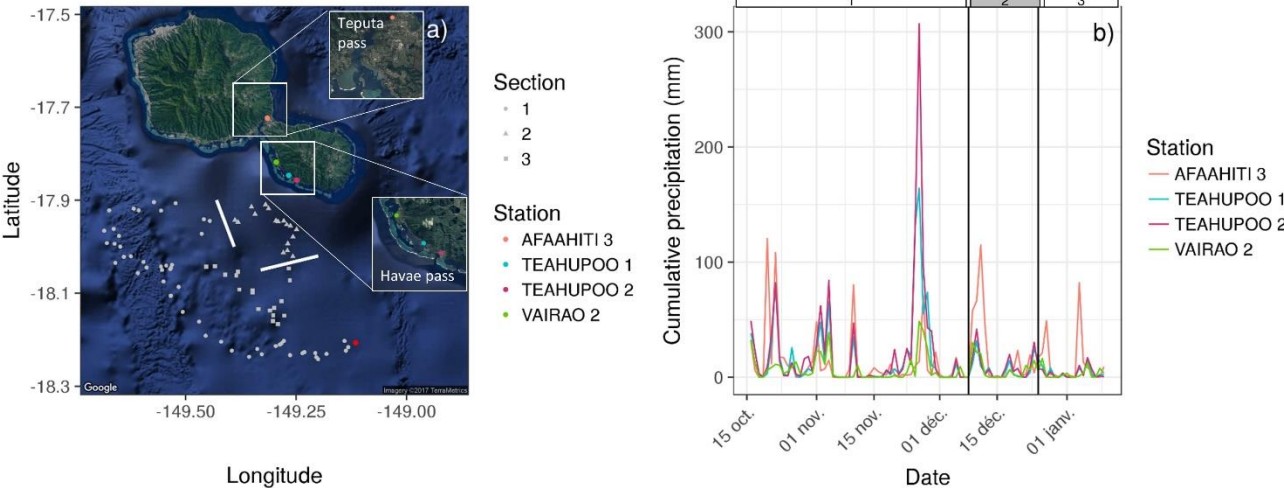

**Figure 7**: **Meteorological data during FTWake lifetime. (a) The FTWake trajectory is plotted as grey points. The red point indicates the location of the float deployment. The different time periods, as defined in Figure 5, are delimited by the white bars along the trajectory of the float. The locations of the 4 Météo-France meteorological stations are detailed in the different insets. (b) Time series of daily cumulative precipitations measured at the 4 stations localized on panel (a). The black vertical solid lines indicate the three specific time periods.**



**Figure 8**: **Time series of the 0 - 300 m HYCOM-modelled current along the FTWake track: (a) current amplitude, (b) zonal and (c) meridian components (units are in m s⁻¹). In panel (b) blue colors represent westward currents while red colors represent eastward currents. In panel (c) blue colors represent southward currents while red colors represent northward currents. The black vertical solid lines in each panel indicate the 3 specific time periods.**





**Figure 9: (a) vertical distribution of FTWake O$_2$ (µmol kg$^{-1}$) along time. Isolines are indicated as black lines (interval=5 µmol kg$^{-1}$). (b) FTWake O$_2$ averaged over 300 m ± 10 m. The black vertical solid lines in each panel indicate the three specific time periods.**





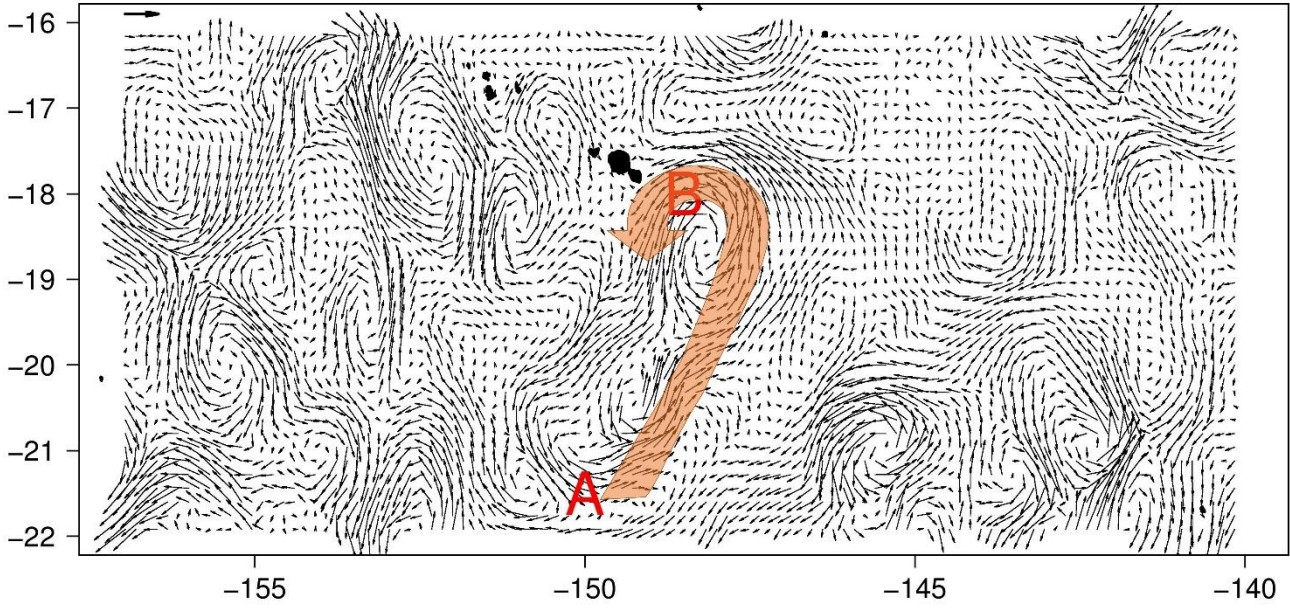

**Figure 10: HYCOM modelled current (in m s$^{-1}$) at 300 m depth for the 04/12/2015, 24 days before the incursion of the more oxygenated waters measured from FTWake. The reference arrow (in bold) is in the top left of the panel and represents 0.5 m s$^{-1}$. The arrow and the points A and B are referring to Section 3.3.2.**





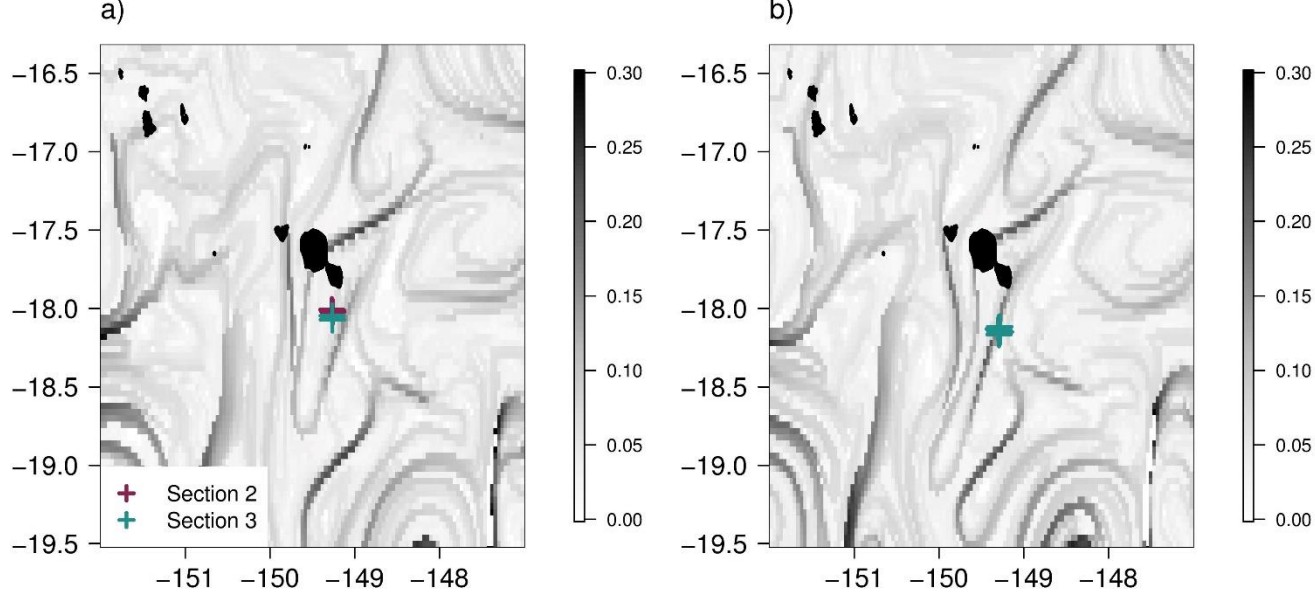

**Figure 11: Spatial distribution of Finite Size Lyapunov Exponents (FSLEs, d⁻¹) represented by the grey color bar for the four days around the (c) 24/12/2015 and (d) 28/12/2015. Crosses in panels (a) and (b) represent the FTWake position with colors referring to the different time periods they belong to. For each map, several crosses are plotted since FSLEs are plotted for four days while 5 FTWake has a daily temporal resolution of data acquisition.**