# Peer review of "Seasonal dynamics and disturbance of phytoplankton biomass in the wake of Tahiti as observed by Biogeochemical-Argo floats"

_Biogeosciences, 2017_

## Referee Comment (RC1) · Anonymous Referee #1 · 2 Feb 2018

This is an interesting paper that looks at the chlorophyll dynamics around Tahiti using primarily data from two BGC-floats. While the paper is publishable, it is not in its current state.

Major issues

1) They downplay the utility of satellite studies of the Island Mass Effect (IME), as they can't resolve subsurface dynamics, yet there is a crucial thing that satellite studies can do, and what is missing from their paper, and that is supply the big picture view of the spatial chlorophyll distribution. They spend a lot of time talking about the chlorophyll dynamics in the "Tahiti wake" but there is no clear idea of what area might be

encompassed by the wake. Furthermore, what dynamic is really going on – is it the Tahiti wake, or the Tahiti IME? They use these terms interchangeable but the two terms imply different things. The Gove et al [2009] paper treats the IME as a chlorophyll increase more or less uniformly distributed around the island, while a wake implies an increase on the downstream side of an island. The two dynamics have different scales as well. The Gove et al [2009] shows the IME extends to only 20-30 km from the coast, whereas chlorophyll increases from wake effects can be seen for hundreds of kms [Andrade et al, 2014]. Which is happening with Tahiti? The climatological chlorophyll image shown in Figure 1 doesn't help with this. They claim the cloudy conditions in the region (pg 12, lines 3-5) preclude the use of satellite data however this is not true. As seen on the attached monthly composites of chlorophyll for 2015, the chlorophyll distribution around Tahiti can be visualized throughout the year. What these figures show is a regional chlorophyll enhancement around Tahiti, not a local one. There are three months, Jun-Aug, ie austral winter, when the satellite chlorophyll levels are markedly higher on one side of the island, however this occurs on the NE side of Tahiti, not on the lee side as the discuss in the paper. Their float data needs to be interpreted in the context of the larger-scale information available from satellite chlorophyll.

2) In the comparison between FOPenO and FTWake in Figure 3 why are just the surface measurements shown? Particularly when the whole point of the BGC floats is to get subsurface data? Also why is the comparison shown before any of the data from FTWake is shown? Figure 5, the sections from FTWake, should directly follow Figure 3. And why isn't the same information shown (as sections) for the two floats? For FOpenO only sections of chlorophyll and density are shown. There is quite a bit of discussion about the procedure for processing the backscatter data but as far as I can tell this data is only shown as depth-averaged values in Fig 3. Likewise for the PAR data.

3) The primary objective of this paper is to examine dynamics in the Tahiti wake, and their primary source of data is from two BGC-Argo floats. However the one from the

wake area is only three months long, so it prohibits examining these dynamics on a seasonal scale. The short length of this float is glossed over in the paper, and never explained why it is so short in duration. They need to be upfront about this shortfall.

4) There are a number of grammatical errors. None of them are major, but there are a lot of them. I have noted some of them but the list is by no means exhaustive. They should have the manuscript edited by a native English speaker.

Point-by-point Comments

Page 1, lines 23-24: "observations collected with two Biogeochemical-Argo (BGC-Argo) profiling floats from April 2015 to November 2016 This implies that they are using data from two floats that both collected data from Apr 2015-Nov 2016, but in fact only one of the floats did. The wake float only collected data for three months. Why is that? This is not explained in the manuscript. The short length of this float is glossed over in the manuscript.

Page 1, lines 27-28: "Vertical observations show a light-driven deepening of the deep chlorophyll maximum (DCM) from winter to summer" This was not shown in the paper. The only PAR data shown was in Figure 3 where it was averaged within the MLD, and there was no representation of the DCM plotted in Figure 3. Its puzzling that they stress in the Intro the importance of their study of having PAR data and then they do not show all the data.

Page 2, line 27: unclear which "this study" is referring to – just specify Gove paper again otherwise the reader might think you are referring to your own paper.

Page 1, line 30-31: "the physical mechanisms involved in the disturbance of phyto-plankton seasonal cycle in the Tahitian wake have been investigated" Since the float in the wake only collected three months of data it is not possible to look at an entire seasonal cycle from the wake, as they claim here.

Page 1, line 35: "bio-optical measurements suggest that the composition of phyto-

plankton community could differ in the Tahitian wake vs. the open ocean area." How so?

Page 3, line 10: unclear which "this study" is referring to – just specify M2014 paper again otherwise the reader might think you are referring to your own paper.

Page 3, line 11 "Indeed M2014 only covered one year of measurements" Careful here – you shouldn't criticize this study too much as a year of data is much more than the FTWake float that forms a central part of your study.

Page 3, line 19-20: "These two pathways allowed the observation of phytoplankton biomass dynamics over a broad range of scales from seasonal to shorter time scales" Again, this falsely leads the reader to assume that seasonal time-scale can be observed for both floats, when in fact this is only the case for the open ocean float. It should be mentioned here that the FTWake float only lasted a few months.

Page 6, line 20-21: "very different drifts experienced by each float allow addressing a broad range of spatial and temporal scales" There is a bit of an overstatement since there is only a three month overlap in the two floats. It's never explained anywhere why FTWake only lasted three months.

Page 7, line 4: The moving average filtering bit should be in the methods, not in the results. The black lines referred to on Figure 3 are barely visible on the figure. What is the significance of the DCM being correlated with the isolumes of chlorophyll? This seems pretty intuitive and a strange way to start the discussion section.

Page 11, lines 14-15: "The open ocean observations confirm the only previous study describing the seasonal vertical dynamics of phytoplankton biomass in the eastern..." You can confirm the results of a previous study, but not the study itself. But what are the results that have been confirmed here?

Tables and Figures Table 2. The asterisk on bbp* is not easily noticed (see my comment below about Figure 3), and it is not an intuitive representation of bbp/chl. Why

not just refer to it as bbp/chl? I think you mean to say Potential Temperature, not Conservative Temperature.

Figure 1. Rather than use the climatological chlorophyll as the background in Figure 1 it would be better to use one of the monthly average during the FTWake float (see attached figures). Climatological distributions show situations that statistically never actually occur. Since this paper focuses primarily on the roughly three month time period that FTWake float was active it would be much more instructive to show conditions during that time period. Also It is hard to interpret the different symbols. Color the two floats different colors, not colored by time, and indicate a few time markers along the trackline of the FOpenO. Also indicate the time period of FOpenO that corresponds to the time period of the FTWake deployment.

Fig. 3. The y-axes of d) and e) have the same variable, but with different units

Figures 2,5 8 and 9. Label the color bars with the variable they are depicting. The black lines referred to in the text for 3a are barely visible. It would be easier to interpret if the months on the x-axis were labeled with month names rather than numbers (ie May not 05) Why is Fig. 9, Oxygen from FTWake separated from the other FTWake sections in Figure 5? It would be much easier to follow the manuscript if all the data from FOpenO was shown together, followed by all the data from FTWake.

Figure 6. What is the point of this figure? What is it telling us about the dynamics of the region?

Figure 8. This figure is hard to interpret. Could you show this information on a map instead? Show vectors of the average surface current on the trackline? The vectors could be three different colors corresponding to period 1, 2 or 3.

Figure 9. Show the MLD on the figure.

Typographical errors

*Pg 1, line 25: change to "The first float transited more than 1000 km"

*Pg 1, line 26: island coast, not island coasts

*Pg 1, line 28: consistent, not consistently

*Pg 1, line 29: change "At the opposite" to "In contrast"

*Pg 1, line 33 (and many other occurrences): precipitation, not precipitations

*Pg 1, line 33: leeward of Tahiti

*Pg 2, line 2: information into the water column

*Pg 2, line 2: can, not could

*Pg 2, line 16: uncertainities in (not to)

*Pg 2, line 19: enhances the (remove the "to")

*Pg 2, line 27: limited to 20°S (remove "the")

*Pg 2, line 27: remove geographical zone

*Page 3, line 4 (and in other place): remove all uses of the "so-called" descriptive, as its use can cast doubt on the authenticity of the term is it be used on.

*Page 6, line 11: used not investigate

*Figure Caption 9 and 10: no "the" before dates, ie should be 300 m depth for 04/12/2015. Same with specific time period – there should be no "the" before period 1 or period 2 etc (page 10, lines 5 and 24)

References

* Bell, and the first Johnson et al. references are missing the journal information.

* Lomas reference title should not be all capitalized.

* Double-check all references for the correct syntax, and make sure extraneous information isn't in them

[Figure]

[Figure]

**Fig. 1.**

---

## Referee Comment (RC2) · Anonymous Referee #2 · 9 Mar 2018

This paper analyzes data from two Biogeochemical-Argo floats deployed from the southwestern side of Tahiti during which one float persisted and drifted westward for approximately 1200 km from April 2015 through November 2016 and the other remained within 45 km of the Tahitian coast for a brief period from October 2015 into January 2016. The paper incorporates meteorological data from the Tahitian coast, model output from the Hybrid Coordinate Ocean Model, and remotely sensed altimetry to aid in interpretation of the observations. This is a valuable complement to observations and the results provide new insights into a generally sparsely sampled portion of the world ocean. However, I believe that the results as presented are not yet up to the high-quality standards required for publication in Biogeoscience. Thus I suggest that

some significant revisions are needed before the article is ready for acceptance. As presented, the authors primarily describe the importance of the seasonal dynamics in the open ocean far field west of Tahiti and distribution of phytoplankton biomass in the wake of Tahiti. The 3 months of in-situ data from the Tahitian wake are insufficient to adequately resolve seasonality of the phytoplankton biomass in the immediate wake and it is not clear that this is necessarily more than one season, or perhaps a transition between seasons. Improvements in presentation and some restructuring could improve the manuscript to a level where it could be acceptable for publication.

The scientific objective of the manuscript is to investigate the seasonal dynamics in phytoplankton biomass in the open ocean compared with observations from the Tahitian wake. The paper's title indicates that the paper will provide insight into the seasonal dynamics and disturbance of phytoplankton biomass of the Tahitian wake. However, the float near the island was out for only 87 days with only 10 of those in the operationally defined summer period limiting the ability to statistically assess and differentiate the summer period from the spring period. Thus, I don't think that the data set for the island wake allows for true resolution of seasonal variability. Although the authors show one year's data for the open ocean float, this data set is not fully exploited nor explained. I suggest the authors consider a title that better represents the results and conclusions of the paper.

Bellow, I list some suggestions to include in the revised manuscript.

1. Given the typical time span of Biogeochemical Argo floats, it would be helpful to explain the short duration of the island wake float. Was this by design, and if so, what is the design criteria? Or was it due to a failure in the float? It would help put the data set into context and understand the limited time duration, given the 19 months of the open ocean deployment.

2. In section 3.1: The authors apply a moving average of +/-5 observations (total of 11 observations/average) in time. Because the open water float's profiling frequency is

shifted from once per day to once every 5 days after 16/07/2015, the averaging period shifts from a 10-day (11 points) average to a 50-day average after 16/07/2015. The implications of this smoothing are much different before and after the breakpoint. The authors in this section describe the dynamic of phytoplankton biomass in the central SPSG. For that purpose, the authors should include the vertical distribution of backscattering in their description, as is an important component to understand the phytoplankton dynamic (in Figure 2). In addition, the authors start the description of Figure 3 about the surface expression of Zpd, MLD, Chlsurf, bbp_surf and their ratio for the open water float without link the data with the vertical distribution (For example integrate CHL0-200m). In that part of the paper, the authors mix the results between the open water float and Tahiti float in Table 3.

3. The authors also try to use mean and range values for the 4 seasons (Table 3), however the data for each season do not have the same temporal distribution and number of observations. I would suggest the authors try to structure the data in Table 3 per month to investigate the mean values of CHL as they did in Figure 3 so you can have a better understanding the phytoplankton biomass distribution. In addition, I would like to suggest the authors rewrite this part, as it is hard for the reader to follow.

4. The authors try to compare the SPSG and Tahitian wake using in-situ data in section 3.2. In the introduction part, the authors point out the importance of subsurface measurements. However, they are using only the surface data to make the comparison between the two study areas. I recommend the authors to restructure this part of the paper and include the vertical distribution of the exam parameters in this sub-section. The authors must show a comparison of the vertical distribution along time of the exam parameters. Several times the authors generalize and compare the spring and summer seasons from both seasons. The Tahiti float has two months of data in the spring and only 10 days of data for the summer season as they call it. In addition, the authors mention in that section that phytoplankton biomass is not visible from remote sensing but both monthly and 8-days products are available (MODIS or CCI). Using the remote

sensing ocean color they will be able to understand better the surfaces changes of phytoplankton dynamics in space and also to compare with the surface in-situ data from the floats. In the last paragraph of this section, the authors present the monthly mean vertical profiles. However, the authors show data only for the 12 months of the open water float without mention and explain the reason? Do the authors find differences in the monthly mean vertical profiles between 2015 and 2016 for the open water float?

5. The authors should explain and justify if they talking about the phytoplankton dynamic in the Tahiti wake or Tahiti island mass effect in section 3.3.1. It is unclear and confusing for the reader to follow up as the change these terms so many times in the manuscript. Furthermore, the authors should answer how the IME effect is relative to the Tahiti Wake as different spatial-temporal processes occur?

6. The discussion in section 3.3.2 is unsatisfactory to me. Nitrate concentrations appear to be elevated in the upper 100 m, in some cases with uniformly high concentrations (>1 $\mu$M) from the surface to 100m, especially in the middle of the period. Rather than showing the T-S plot which is interesting, the actual vertical profiles of temperature an salinity would be helpful. Density, as the authors indicate, shows stratification in the upper layer which is consistent either with the possible or upwelling or land runoff of fresh water which will be buoyant and should only be reflected in the near-surface region.

7. Section 3.3.2 – Lines 16-23 – you suggest possible entrainment from a lagoon driven by a strong current around the island. It is interesting that the average O2 concentration in the upper 300 m during this period declines and there is clearly decreased oxygen and increased nitrate deeper in the water column (Figures 5e and 9). Do you know anything about the AOU/Nitrate relationship for the region? Is there any possibility that some sort of mixing or upwelling (upstream) might have brought this about?

8. The differential in the water masses based on your T-S diagrams might give some insight. You code the T-s plot with chlorophyll. It could be useful to look at this with

respect to oxygen to try to understand the sources of nutrients during period 2.

9. The speculation using the Hycom model is a useful exercise, but how confident are you in the results? Is the model assimilating the regional float data and regional altimetry?

10. The authors in the summary again refer to IME effect using data from Tahiti Wake float. Did the authors examine remotely sensed chlorophyll and SSH to provide further characterization of the region? Perhaps there could be some clues

Other comments

Authors have to harmonize the abbreviations in the document

In Section 3.2, Line 4-5 the following statement is made – "likely reflect differences in phytoplankton community composition (or in the nature of the particle assemblage) in the island wake as compared to the open ocean." Besides bbp*, is there any other data like HPLC data from profiles near the Wake float that provide additional insight?

Figure 2: Units and labels are missing. Isolumes lines are not visible in the CHL plot (panel a) in the PDF submitted for review.

Figure 5: Units and label are missing from the figure from c-h.

In the legend of the figure 5e and 5f the authors refer that the zoom is in the first 150m but in the plot shows only the first 100m of the water column

In figure 5b the top boxes that indicate the periods are not in same locations as in the others subplots

It will be useful for the reader if the authors can add the MLD in figure 5e and 5f

Are you confident in the nitrate concentrations shown in Figure 5? During late October – early November, you show nitrate concentrations on the order of 0.5-0.8 $\mu$M after the water column has stratified. And the concentrations seem to be uniform throughout the

upper 100 m. These values seem high for an oligotrophic sea, but I have no experience or literature understanding of this region.

Figure 7. The section symbols are small and hard to identify on the map

Figure 8 Units and labels are missing

Figure 9: The authors should show the MLD and the DCM. Also, I suggest the authors show all the data from the Tahiti Wake float together.

---

## Author Comment (AC1) · 21 Apr 2018

First of all, we are deeply grateful to Reviewer 1 for his/her constructive comments and suggestions to improve our manuscript. Here we address in details and point-by-point these comments. Our responses follow each comment in blue.

Answers to major issues raised by Reviewer 1 (mentioned as R1 hereafter):

**R1's point 1a:** They downplay the utility of satellite studies of the Island Mass Effect (IME), as they can't resolve subsurface dynamics, yet there is a crucial thing that satellite studies can do, and what is missing from their paper, and that is supply the big picture view of the spatial chlorophyll distribution. They spend a lot of time talking about the chlorophyll dynamics in the "Tahiti wake" but there is no clear idea of what area might be encompassed by the wake. Furthermore, what dynamic is really going on – is it the Tahiti wake, or the Tahiti IME? They use these terms interchangeable but the two terms imply different things. The Gove et al [2009] paper treats the IME as a chlorophyll increase more or less uniformally distributed around the island, while a wake implies an increase on the downstream side of an island. The two dynamics have different scales as well. The Gove et al [2009] shows the IME extends to only 20-30 km from the coast, whereas chlorophyll increases from wake effects can be seen for hundreds of kms [Andrade et al, 2014]. Which is happening with Tahiti?

Author's response (mentioned as AR hereafter in this document) to point 1a: We agree with R1 that satellite ChI data can supply the big picture view of the spatial ChI distribution. However, we have examined the monthly and 8-day composites of ChI satellite data from 2002 to 2017 in the studied area but we have not seen any clear biological enhancement leeward of Tahiti from these data (see in addition the AR to point 1b and 1c below).

In the manuscript, we used the term "Tahitian wake" to refer to the geographic position of the float leeward of the island, not to the physical process. The term "IME" refers to the biological enhancement induced by the island, as defined by Doty and Ogury (1956). Considering the R1 and R2's comments, we agree that there can be some confusion with this terminology. To avoid any misunderstanding, we removed the term "island wake" used in the manuscript and replaced it with "leeward of Tahiti". The FTWake float is now referred to as FLeeT float.

R1's point 1b: The climatological chlorophyll image shown in Figure 1 doesn't help

**BGD**
with this. They claim the cloudy conditions in the region (pg 12, lines 3-5) preclude the use of satellite data however this is not true. As seen on the attached monthly composites of chlorophyll for 2015, the chlorophyll distribution around Tahiti can be visualized throughout the year.

**AR's point 1b:** We agree with R1 that we were not clear enough on this point. The mention of cloudy conditions preventing the use of ocean color satellite observations in the text specifically referred to the study of the IME period of interest (i.e., during  $\sim 15$  days in December 2015). Considering the short time scales of this phenomenon, the use of monthly composites is not adapted to follow the dynamics and the evolution of the IME. The 8-days composites have a too sparse spatio-temporal coverage as only the composite covering the 11th to the 19th of December period shows some data leeward of Tahiti. The other three 8-day composites in December are extremely cloudy. We agree that we were not clear enough on this point. The sentence has been consequently modified and completed.

According to the R1's suggestion, we have also substituted the previous annual climatological satellite image in Figure 1 with the OC-CCI monthly composite of December 2015 (when the IME is observed). Moreover, we have split this figure in 3 sub-parts (one at basin-scale, one centered on FOpenO and one zoomed on the FLeeT trajectory) to provide a more exhaustive view of the studied area (see the new Figure 1 below).

In section 3.2 of the revised manuscript, we propose to add the following sentence: "The OC-CCI monthly composite in December 2015 (comprising the biological enhancement event observed in Figure 3) shows high ChI ( $\sim 0.8 \ mg \ m^{-3}$ ) eastward of the FLeeT trajectory and nearby the southern coast of Tahiti (Figure 1c). This imprint on ocean color satellite observations might be linked to the biological enhancement highlighted in Figure 3. However, the use of monthly composites is not adapted to follow the dynamics and the time evolution of this event and the 8-days composites have a too sparse spatio-temporal coverage due to a dense cloud cover."

**BGD**
**R1's point 1c:** What these figures show is a regional chlorophyll enhancement around Tahiti, not a local one. There are three months, Jun-Aug, ie austral winter, when the satellite chlorophyll levels are markedly higher on one side of the island, however this occurs on the NE side of Tahiti, not on the lee side as the discuss in the paper. Their float data needs to be interpreted in the context of the larger-scale information available from satellite chlorophyll.

**AR's point 1c:** We agree with R1's suggestion and consequently, the basin-scale ChI satellite data has been added to Figure 1 (see Fig. 1a below) to provide a more exhaustive view of the ChI spatial distribution. On this new Figure 1a, Tahiti is located in the upper-right corner of the white rectangle (see the white point). The general trophic conditions in the South Pacific Subtropical Gyre are shown in this figure with higher ChI in the NE side of Tahiti toward the equatorial mesotrophic region, and lower ChI southwestward toward the oligotrophic gyre. The increase of chlorophyll during winter, pointed out by R1, is consistent with our investigation of the seasonal variability from FOpenO (see Figure 3c).

**R1's point 2a:** In the comparison between FOpenO and FTWake in Figure 3 why are just the surface measurements shown? Particularly when the whole point of the BGC floats is to get subsurface data? Also why is the comparison shown before any of the data from FTWake is shown? Figure 5, the sections from FTWake, should directly follow Figure 3.

**AR's point 2a:** We reproduced plots based on surface measurements similar to the one of Mignot et al. 2014 (M2014) in order to compare the central oligotrophic region of the South Pacific Ocean (our studied area, see the white rectangle in the new Figure 1a below) with the eastern ultra-oligotrophic area of M2014 (see the white star in the new Figure 1a below).

Figure 3 a) shows that our results are in agreement with M2014 results in the SPSG, b) provides the open ocean seasonal context of our studied area which then c) allows

**BGD**
us to introduce and highlight the IME from FLeeT that is presented in Figure 5. Hence, we believe that moving Figure 5 directly after Figure 2 would perturb the common thread of the article. However, according to R1's comment, we propose to add some information at the beginning of this section to clarify our strategy and the plan of the manuscript.

In section 3.1 of the revised manuscript, we propose to add the following sentence: "The seasonal dynamics of phytoplankton biomass in the central SPSG is investigated using the 18 months of observations from FOpenO. Our findings are compared with M2014 results to provide a new insight of the seasonal dynamics of phytoplankton biomass in the central region of the South Pacific Ocean which is less oligotrophic than the eastern ultra-oligotrophic area of their study (see the white rectangle and the white star in Figure 1a respectively)."

**R1's point 2b:** And why isn't the same information shown (as sections) for the two floats? For FOpenO only sections of chlorophyll and density are shown. **AR's point 2b:** Information shown in Figures 2 and 5 differ because FOpenO does not have some nitrate and oxygen sensors. Chl as a function of density (Figure 5g) aims to illustrate that the vertical distribution of Chl is disturbed during period 2 (IME). This plot does not add any more information for FOpenO seasonal dynamics.

**R1's point 2c:** There is quite a bit of discussion about the procedure for processing the backscatter data but as far as I can tell this data is only shown as depth-averaged values in Fig 3. Likewise for the PAR data.

**AR's point 2c:** We would like to point out that the PAR and  $b_{bp}$  data are also used in Figures 2, 4 and 5. The PAR data are represented as isolumes in Figures 2a and 5a while the  $b_{bp}$  is represented in Figure 4 and in Figure 5a. Hence, we think that it is
important to well describe the processing of these data.

According to R1 and R2's suggestions, we will add the time series of Particulate Organic Carbon (POC) that is derived from  $b_{bp}$  and PAR in Figures 2 and 5 of the revised manuscript. We also have transformed every  $b_{bp}$  values into POC values in order to make it more representative to the reader (for example in Figure 4). The POC conversion is already explained in the manuscript (lines 25-33 page 4 and lines 1-3 page 5 of the initial manuscript).

**R1's point 3:** The primary objective of this paper is to examine dynamics in the Tahiti wake, and their primary source of data is from two BGC-Argo floats. However the one from the wake area is only three months long, so it prohibits examining these dynamics on a seasonal scale. The short length of this float is glossed over in the paper, and never explained why it is so short in duration. They need to be upfront about this shortfall.

**AR's point 3:** We agree that the seasonal dynamics in the phytoplankton biomass leeward of Tahiti cannot be fully investigated with our 3 months-data. We thought that the reason of the short length of this float was implicit. The float stopped communicating after three months and was declared lost and inactive. A sentence has been added to clarify this point in section 2.1: "Because of a technical issue, the FLeeT stopped communicating after 3 months of data acquisition, limiting our study to only 3 months of data leeward of Tahiti."

**R1's point 4:** There are a number of grammatical errors. None of them are major, but there are a lot of them. I have noted some of them but the list is by no means exhaustive. They should have the manuscript edited by a native English speaker. **AR's point 4:** The revised manuscript will be corrected by a native English speaker.

BGD
Answers to R1's point-by-point minor comments:

**R1:** Page 1, lines 23-24: "observations collected with two Biogeochemical-Argo (BGCArgo) profiling floats from April 2015 to November 2016. This implies that they are using data from two floats that both collected data from Apr 2015-Nov 2016, but in fact only one of the floats did. The wake float only collected data for three months. Why is that? This is not explained in the manuscript. The short length of this float is glossed over in the manuscript.

**AR:** We will modify the sentence to be more precise as follows: "Physical and biogeochemical observations collected with two Biogeochemical-Argo (BGC-Argo) profiling floats are used to investigate the dynamics of phytoplankton biomass. The first float drifted from April 2015 to November 2016 over more than 1000 km westward of Tahiti (open ocean conditions). The second float was deployed leeward of Tahiti in October 2015 and stopped communicating in January 2016. Over these 4 months, it remained within 45 km off the island coast."

**R1:** Page 1, lines 27-28: "Vertical observations show a light-driven deepening of the deep chlorophyll maximum (DCM) from winter to summer" This was not shown in the paper. The only PAR data shown was in Figure 3 where it was averaged within the MLD, and there was no representation of the DCM plotted in Figure 3. Its puzzling that they stress in the Intro the importance of their study of having PAR data and then they do not show all the data.

**AR:** The seasonal variability along depth of the PAR is shown in Figure 2a (Section 3.1). The 0.1, 1 and 10 *mol* photons  $m^{-2} d^{-1}$  isolumes are represented as dotted
black lines while the DCM depth is represented by the red line in Figure 2a. However, we agree that the black dotted lines were barely visible in this figure so we added the PAR time series to Figure 2 (see the new Figure 2 below). In addition, the importance of the PAR is highlighted through the correlation between the DCM depth and the 1 *mol photons*  $m^{-2} d^{-1}$  isolume: "the depths of the DCM and of the specific isolume of 1 *mol photons*  $m^{-2} d^{-1}$  are significantly correlated (r = 0.89 and p-value

for instance Cetinic et al., 2012). This is shown and discussed in Section 3.2 of the initial manuscript (lines 3-14 page 8). To be clearer, the sentence will be completed as follow: "Moreover, a bio-optical-based community index suggests that the composition of phytoplankton community could differ leeward of Tahiti vs. the open ocean area."

**R1:** Page 3, line 10: unclear which "this study" is referring to – just specify M2014 paper again otherwise the reader might think you are referring to your own paper. **AR:** The sentence has been changed as suggested.

**R1:** Page 3, line 11: "Indeed M2014 only covered one year of measurements" Careful here – you shouldn't criticize this study too much as a year of data is much more than the FTWake float that forms a central part of your study.

**AR:** We agree with R1 that the sentence was awkward as our aim was to highlight our opportunity to investigate an additional year and a half of BGC-Argo observations to the sole year of data described so far. The sentence has been modified. Moreover, we have revised the manuscript to be clear that the seasonal cycle is studied only from the open ocean float.

**R1:** Page 3, line 19-20: "These two pathways allowed the observation of phytoplankton biomass dynamics over a broad range of scales from seasonal to shorter time scales". Again, this falsely leads the reader to assume that seasonal time-scale can be observed for both floats, when in fact this is only the case for the open ocean float. It should be mentioned here that the FTWake float only lasted a few months. **AR:** We agree with R1. This sentence has been reformulated to be clear on this point (as anywhere else in the manuscript). BGD
**R1:** Page 6, line 20-21: "very different drifts experienced by each float allow addressing a broad range of spatial and temporal scales" There is a bit of an overstatement since there is only a three month overlap in the two floats. It's never explained anywhere why FTWake only lasted three months.

**AR:** We agree and consequently corrected the manuscript (see comments above).

**R1:** Page 7, line 4: The moving average filtering bit should be in the methods, not in the results. The black lines referred to on Figure 3 are barely visible on the figure. What is the significance of the DCM being correlated with the isolumes of chlorophyll? This seems pretty intuitive and a strange way to start the discussion section.

**AR:** According to R1's suggestion, the moving average description has been displaced to the Method section. We assume that R1 refers here to the black lines on Figure 2 and not Figure 3.

As mentioned above, we have added a panel with the PAR time series to Figure 2 (see the new Fig. 2 below) to better see the isolumes. The correlation of the DCM depth to isolumes (PAR) highlights that the DCM depth is light driven. This result confirms, for the oligotrophic central Pacific in 2015/2016, what Mignot et al. (2014) reported in the ultra-oligotrophic area in 2009. This result is not so intuitive as the light-driven seasonal dynamics of chlorophyll was previously described only by Letelier et al. (2004) and Mignot et al. (2014) for other oligotrophic environments.

**R1:** Page 11, lines 14-15: "The open ocean observations confirm the only previous study describing the seasonal vertical dynamics of phytoplankton biomass in the eastern:" You can confirm the results of a previous study, but not the study itself. But
what are the results that have been confirmed here?

**AR:** Here we have confirmed 1) the light-driven deepening from winter to summer of the DCM depth and 2) that the ChI wintertime increase in the upper layer is likely due to photoacclimation process. According to R1's comment, we have reformulated the sentence as follow: "Using in situ PAR measurements acquired from FOpenO float (vs. PAR derived from downward irradiance in M2014), we confirm that the wintertime ChI increase in the upper layer is likely due to photoacclimation and that the seasonal DCM vertical variability is light-driven in the present open ocean oligotrophic environment."

Answers to R1's comments on "Tables and Figures":

**R1:** Table 2. The asterisk on  $b_{bp}^*$  is not easily noticed (see my comment below about Figure 3), and it is not an intuitive representation of  $b_{bp}$ /chl. Why not just refer to it as  $b_{bp}$ /chl? I think you mean to say Potential Temperature, not Conservative Temperature. **AR:** As suggested by R1, we have replaced  $b_{bp}^*$  by  $b_{bp}$ /Chl. We really mean Conservative Temperature that is the new standard for ocean temperature following the TEOS-10 procedures as described by McDougall et al. (2011, 2012).

**R1:** Figure 1. Rather than use the climatological chlorophyll as the background in Figure 1 it would be better to use one of the monthly average during the FTWake float (see attached figures). Climatological distributions show situations that statistically never actually occur. Since this paper focuses primarily on the roughly three month time period that FTWake float was active it would be much more instructive to show conditions during that time period. Also it is hard to interpret the different symbols. Color the two floats different colors, not colored by time, and indicate a few time

BGD
markers along the trackline of the FOpenO. Also indicate the time period of FOpenO that corresponds to the time period of the FTWake deployment.

**AR:** We modified Figure 1 as suggested by R1 (see Fig. 1 below). Moreover, we have split this figure in 3 sub-parts (one at basin-scale, one centered on FOpenO and one zoomed on the FLeeT trajectory). FOpenO profiles concomitant with FLeeT profiles acquisition are colored in red.

**R1:** Figure 3. The y-axes of d) and e) have the same variable, but with different units **AR:** In Figure 3, the y-axis in (d) represents surface  $b_{bp}$  whereas the y-axis in (e) represents the surface  $b_{bp}^*$ . As mentioned above, to avoid confusion, we have changed  $b_{bp}^*$  in  $b_{bp}$ /Chl.

**R1:** Figures 2, 5, 8 and 9. Label the color bars with the variable they are depicting. The black lines referred to in the text for 3a are barely visible. It would be easier to interpret if the months on the x-axis were labeled with month names rather than numbers (ie May not 05). Why is Fig. 9, Oxygen from FTWake separated from the other FTWake sections in Figure 5? It would be much easier to follow the manuscript if all the data from FOpenO was shown together, followed by all the data from FTWake. **AR:** Figures 2, 5, 8 and 9 have been modified as suggested. We believe that the common thread of the article is easier to follow keeping Figure 9 separately from Figure 5. Indeed, Figure 5 mostly highlights biogeochemical changes from period 1 to 2 and 3 over the 150-200 upper meters. It allows us to start the discussion on rain events and island run off as a mechanism to explain changes from period 1 to 2 (Figures 6 and 7). Then, Figure 8 gives insights on the transition from period 2 to 3 through dynamical changes over 0-300 m, which are finally further investigated in Figure 9 and later in the manuscript. Hence, we believe that moving all together the
FOpenO figures, followed by all the FLeeT Figures would perturb the common thread of our manuscript. However, according to R1's comment, we propose to add some information at the beginning of this section to clarify our strategy and the plan of the manuscript.

**R1:** Figure 6. What is the point of this figure? What is it telling us about the dynamics of the region?

**AR:** The point of this figure is to show 1) the relatively low sea surface salinity over periods 2 and 3 likely induced by rain events and 2) that a change in water mass can be evidenced from period 2 to 3. According to a comment of the second reviewer, we have colored the T-S diagram with respect to oxygen and nitrate concentrations instead of Chl in the revised version to provide insights on nitrate and oxygen sources in the lower layer, which is of interest when considering the transition from period 2 to 3 (see Fig. 3 below).

**R1:** Figure 8. This figure is hard to interpret. Could you show this information on a map instead? Show vectors of the average surface current on the trackline? The vectors could be three different colors corresponding to period 1, 2 or 3.

**AR:** The vertical distribution cannot be presented on a map and we think that representing the average surface current would remove some interesting information discussed in the manuscript such as the change in the ocean dynamics between the periods 2 to 3 down to 300 m (lines 25-27 and lines 30-31 page 10 of the initial manuscript). Therefore, we prefer to keep this figure as in the initial version.

R1: Figure 9. Show the MLD on the figure.
Finally, all typographical errors were corrected according to R1's suggestions. We checked all references for the correct syntax.

Legends of the attached figures below :

**Fig.1 (new Figure 1 in the revised manuscript):** (a) Spatial distribution of OC-CCI surface satellite ChI ( $mg \ m^{-3}$ ) in December 2015 for the Pacific Ocean. Our studied area and the Tahiti island are represented by the white rectangle and the white point respectively. The white star represents the geographic area of the M2014 reference study (see Section 3.1). The trajectories of (b) FOpenO float and (c) FLeeT float are shown with color in background representing the OC-CCI surface satellite ChI ( $mg \ m^{-3}$ ) distribution in December 2015. In panel b FOpenO profiles concomitant with FLeeT profiles acquisition are colored in red. Grey pixels represent missing data because of clouds or the 500 m bathymetric mask that removed nearshore pixels that could be biased by land. Islands and continents are indicated in black. The Moorea and Tahiti islands are indicated and the mean direction of the South Equatorial Current (SEC) is indicated by the arrow in panel c.

**Fig. 2 (new Figure 2 in the revised manuscript):** FOpenO vertical distribution along time of (a) ChI ( $mg \ m^{-3}$ ) with the 0.1, 0.2 and 0.3  $mg \ m^{-3}$  isocontours as dotted black lines, (b) PAR ( $mol \ photons \ m^{-2} \ d^{-1}$ ) with the the 0.1, 1 and 10  $mol \ photons \ m^{-2} \ d^{-1}$  isolumes as dotted black lines, (c) POC ( $mg \ m^{-3}$ ) and (d) density ( $kg \ m^{-3}$ ) with isopy-cnes (interval = 0.5  $kg \ m^{-3}$ , dotted black lines). In each panel, the white and red lines
show the MLD and the depth of the DCM, respectively.

**Fig. 3 (new Figure 6 in the revised manuscript):** T-S diagrams issued from FLeeT during period 2 (a and c panels) and period 3 (b and d panels) as defined in the text (see Section 3.3.1) and Figure 5. Black dotted lines represent isopycnal surfaces (interval= 0.5  $kg m^{-3}$ ). As shown in color, T-S measurements are associated with O2 concentrations ( $mol kg^{-3}$ , a and b panels) and NO3- concentrations ( $mol kg^{-3}$ , c and d panels).

**BGD**

---

## Author Comment (AC2) · 21 Apr 2018

First of all, we are deeply grateful to Reviewer 2 for his/her constructive comments and suggestions to improve our manuscript. Here we address in details and point-by-point these comments. Our responses follow each comment in blue.

Answers to the general comment of Reviewer 2 (mentioned as R2 hereafter):

[Figure]

**R2:** The scientific objective of the manuscript is to investigate the seasonal dynamics in phytoplankton biomass in the open ocean compared with observations from the Tahitian wake. The paper's title indicates that the paper will provide insight into the seasonal dynamics and disturbance of phytoplankton biomass of the Tahitian wake. However, the float near the island was out for only 87 days with only 10 of those in the operationally defined summer period limiting the ability to statistically assess and differentiate the summer period from the spring period. Thus, I don't think that the data set for the island wake allows for true resolution of seasonal variability. Although the authors show one year's data for the open ocean float, this data set is not fully exploited nor explained. I suggest the authors consider a title that better represents the results and conclusions of the paper.

**Author's response (mentioned as AR hereafter):** We proposed our previous title in order to highlight: 1) the seasonal dynamics in the open ocean (from FOpenO) and 2) the signature of an IME on phytoplankton biomass (from FTWake). However, we agree that the title was a little bit misleading as the seasonal dynamics in the phytoplankton biomass leeward of Tahiti cannot be fully investigated. Hence, in agreement with R2's comment and, in some way with R1's response, we propose another title as: "Enhancement of phytoplankton biomass leeward of Tahiti as observed by Biogeochemical-Argo floats".

Answers to R2's list of suggestions:

**R2's point 1:** Given the typical time span of Biogeochemical Argo floats, it would be helpful to explain the short duration of the island wake float. Was this by design, and if so, what is the design criteria? Or was it due to a failure in the float? It would help put the data set into context and understand the limited time duration, given the 19 months

of the open ocean deployment.

**AR's point 1:** We thought that the reason was implicit. The FTWake float (now FLeeT according to R1's comments and R2's point 5) stopped communicating after three months and was declared lost and inactive. A sentence has been added to clarify this point in section 2.1: "Because of a technical issue, the FLeeT stopped communicating after 3 months of data acquisition, limiting our study to only 3 months of data leeward of Tahiti."

**R2's point 2a:** In section 3.1: The authors apply a moving average of +/-5 observations (total of 11 observations/average) in time. Because the open water float's profiling frequency is shifted from once per day to once every 5 days after 16/07/2015, the averaging period shifts from a 10-day (11 points) average to a 50-day average after 16/07/2015. The implications of this smoothing are much different before and after the breakpoint.

**AR's point 2a:** According to R2's suggestions, we redid the calculation with a moving average of 1) +/- 15 observations (total of 31 points/average) when the float has a daily profiling frequency (i.e., until the 16/07/2016 for FOpenO and during all the lifetime of FLeeT) and 2) +/- 3 observations (total of 7 points/average) when the FOpenO float has a 5 days frequency of acquisition. Hence, a steady moving average period of 30 days is applied for the 2 floats over their lifetime. In the corrected manuscript, this moving average description has been added in the Method section.

**R2's point 2b:** The authors in this section describe the dynamic of phytoplankton biomass in the central SPSG. For that purpose, the authors should include the vertical distribution of backscattering in their description, as is an important component to understand the phytoplankton dynamic (in Figure 2).

**AR's point 2b:** In agreement with R2, we included the time series of POC (linear transformation from $b_{bp}$) in Figure 2 (see the new figure attached as Fig. 1 below). We have also transformed every $b_{bp}$ values in POC values in the revised manuscript in order to make it more representative to the reader (for example in Figure 4). The POC conversion was already explained in the initial manuscript lines 25-33 page 4 and lines 1-3 page 5.

**R2's point 2c:** In addition, the authors start the description of Figure 3 about the surface expression of Zpd, MLD, Chlsurf, bbpsurf and their ratio for the open water float without link the data with the vertical distribution (For example integrate CHL0-200m). In that part of the paper, the authors mix the results between the open water float and Tahiti float in Table 3.

**AR's point 2c:** We reproduced plot based on surface measurements similar to the one of Mignot et al. 2014 (M2014) in order to compare the central oligotrophic region of the South Pacific Ocean (our studied area, see the white rectangle in the new Figure 1, see Fig. 2a below) with the eastern ultra-oligotrophic area of M2014 (see the white star in Figure 2a below).

Figure 3 a) shows that our results are in agreement with M2014 results in the SPSG, b) provides the open ocean seasonal context of our studied area which then c) allows us to introduce and highlight the IME from FLeeT that is presented in Figure 5. Hence, we believe that moving Figure 5 directly after Figure 2 would perturb the common thread of the article. However, according to R2's comment, we propose to add some information at the beginning of this section to clarify our strategy and the plan of the manuscript.

In section 3.1 of the revised manuscript, we propose to add the following sentence: "The seasonal dynamics of phytoplankton biomass in the central SPSG is investigated using the 18 months of observations from FOpenO. Our findings are compared with M2014 results to provide a new insight of the seasonal dynamics of phytoplankton

biomass in the central region of the South Pacific Ocean which is less oligotrophic than the eastern ultra-oligotrophic area of their study (see the white rectangle and the white star in Figure 1a respectively)."
According to R2's comment on Figure 3, we have removed the FLeeT data from Table 3 and created a specific Table 4.

**R2's point 3:** The authors also try to use mean and range values for the 4 seasons (Table 3), however the data for each season do not have the same temporal distribution and number of observations. I would suggest the authors try to structure the data in Table 3 per month to investigate the mean values of CHL as they did in Figure 3 so you can have a better understanding the phytoplankton biomass distribution. In addition, I would like to suggest the authors rewrite this part, as it is hard for the reader to follow.
**AR's point 3:** As suggested by R2, we organized Table 3 per month instead of seasons. This part has been rewritten in the revised manuscript.

**R2's point 4a:** The authors try to compare the SPSG and Tahitian wake using in-situ data in section 3.2. In the introduction part, the authors point out the importance of subsurface measurements. However, they are using only the surface data to make the comparison between the two study areas. I recommend the authors to restructure this part of the paper and include the vertical distribution of the exam parameters in this sub-section. The authors must show a comparison of the vertical distribution along time of the exam parameters. Several times the authors generalize and compare the spring and summer seasons from both seasons. The Tahiti float has two months of data in the spring and only 10 days of data for the summer season as they call it.
**AR's point 4a:** Comparisons between the open ocean and leeward of Tahiti have not been performed on surface data only. The mean vertical distributions of the examined

parameters are also presented and compared in Figure 4.

We agree that we cannot generalize the seasonal dynamics using only the 3-month data from FLeeT. Therefore, we consequently corrected the manuscript.

**R2's point 4b:** In addition, the authors mention in that section that phytoplankton biomass is not visible from remote sensing but both monthly and 8-days products are available (MODIS or CCI). Using the remote sensing ocean color they will be able to understand better the surfaces changes of phytoplankton dynamics in space and also to compare with the surface in-situ data from the floats. In the last paragraph of this section, the authors present the monthly mean vertical profiles. However, the authors show data only for the 12 months of the open water float without mention and explain the reason? Do the authors find differences in the monthly mean vertical profiles between 2015 and 2016 for the open water float?

**AR's point 4b:** We agree with R2 that we were not clear enough on this point. The mention of cloudy conditions preventing the use of ocean color satellite observations in the text specifically referred to the study of the IME period of interest (i.e., during $\sim 15$ days in December 2015). Considering the short time scales of this phenomenon, the use of monthly composites is not adapted to follow the dynamics and the evolution of the IME. The 8-days composites have a too sparse spatio-temporal coverage as only the composite covering the 11th to the 19th of December period shows some data leeward of Tahiti. The other three 8-day composites in December are extremely cloudy. We agree that we were not clear enough on this point. The sentence has been consequently modified and completed.

We have also substituted the previous annual climatological satellite image in Figure 1 with the OC-CCI monthly composite of December 2015 (when the IME is observed). Moreover, we have split this figure in 3 (one at basin-scale, one centered on FOpenO and one zoomed on the FLeeT trajectory) to provide a more exhaustive view of the studied area (see this new figure as Fig. 2 below).

In section 3.2 of the revised manuscript, we propose to add the following sentence: "The OC-CCI monthly composite in December 2015 (comprising the biological enhancement event observed in Figure 3) shows high Chl ($\sim 0.8\ mg\ m^{-3}$) eastward of the FLeeT trajectory and nearby the southern coast of Tahiti (Figure 1c). This imprint on ocean color satellite observations might be linked to the biological enhancement highlighted in Figure 3. However, the use of monthly composites is not adapted to follow the dynamics and the time evolution of this event and the 8-days composites have a too sparse spatio-temporal coverage due to a dense cloud cover."

We agree with R2 that showing FOpenO vertical profiles for 12 months without taking into account years 2015 or 2016 in Figure 4 only provides a partial view. The Figure 4 as a function of years (i.e. 2015 and 2016) is presented below as Fig. 3. There is no significant difference between years 2015 and 2016. Therefore, we think that the figure as in the initial manuscript is easier to follow as years do not bring any more information to that part of the study. However, we added in the text the information that there is no significant difference between years.

**R2's point 5:** The authors should explain and justify if they talking about the phytoplankton dynamic in the Tahiti wake or Tahiti island mass effect in section 3.3.1. It is unclear and confusing for the reader to follow up as the change these terms so many times in the manuscript. Furthermore, the authors should answer how the IME effect is relative to the Tahiti Wake as different spatial-temporal processes occur?

**AR's response to point 5:** In the manuscript, we used the term "Tahitian wake" to refer to the geographic position of the float leeward of the island, not to the physical process. The term "IME" refers to the biological enhancement induced by the island, as defined by Doty and Ogury (1956). Considering the R1 and R2's comments, we agree that there can be some confusion with this terminology. To avoid any misunderstanding, we removed the term "island wake" used in the manuscript and replaced it with "leeward of Tahiti". The FTWake float is now referred to as FLeeT float.

**R2's point 6:** The discussion in section 3.3.2 is unsatisfactory to me. Nitrate con-
centrations appear to be elevated in the upper 100 m, in some cases with uniformly
high concentrations (>1 $\mu$M) from the surface to 100m, especially in the middle of
the period. Rather than showing the T-S plot which is interesting, the actual vertical
profiles of temperature and salinity would be helpful. Density, as the authors indicate,
shows stratification in the upper layer which is consistent either with the possible
or upwelling or land runoff of fresh water which will be buoyant and should only be
reflected in the near-surface region.

**AR's point 6:** Time series of temperature and salinity (for both the 0-350 m and the
0-100 m layers) are shown in Fig. 4 below. On one hand, a salinity decrease can be
observed in the upper layer during period 2, in agreement with land runoff of fresh
water (as in the TS diagram). On the other hand, there is no temperature decrease
in the upper layer or uplift of isotherms possibly associated with an upwelling event.
The upwelling assumption has also been investigated through satellite Sea Surface
Temperature observations during period 2 with no conclusive outcomes. However, we
agree that these information were missing in the text, hence the following sentence
will be added: "Temperature measured from FLeeT remains constant along time and
depth in the upper layer over period 2 (figure not shown). Moreover, sea surface
temperature observations derived from satellite during this period do not show any
cold-water pattern leeward of Tahiti. Hence, the coastal upwelling assumption to
provide nutrients toward the surface has not been retained."

**R2's point 7:** Section 3.3.2 – Lines 16-23 – you suggest possible entrainment from a
lagoon driven by a strong current around the island. It is interesting that the average
O2 concentration in the upper 300 m during this period declines and there is clearly

decreased oxygen and increased nitrate deeper in the water column (Figures 5e and 9). Do you know anything about the AOU/Nitrate relationship for the region? Is there any possibility that some sort of mixing or upwelling (upstream) might have brought this about?

**AR's point 7:** First of all, we would like to bring to the attention of R2 that a) the decline of O2 is observed at a depth of 300 +/-10 m (this is not a 0-300m average that is represented in Figure 9b) and b) Figure 5e is a zoom of Chl not of nitrate concentrations (the increase of nitrate at 300 m can be seen in Figure 5d). As far as we know nothing is reported about the AOU/nitrate relationship for the studied region.

**R2's point 8:** The differential in the water masses based on your T-S diagrams might give some insight. You code the T-s plot with chlorophyll. It could be useful to look at this with respect to oxygen to try to understand the sources of nutrients during period 2.

**AR's point 8:** During the second period, Figure 5 shows an input of nitrate in the upper 100m likely coming from the surface (Figure 5f) and not from an uplift of the deep-rich layers (consistently with the physical mechanisms presented in Figures 5 to 7). However, we agree with R2 that the T-S diagram colored with respect to oxygen and nitrate concentrations (see Fig. 5 below) provide insights on nitrate and oxygen sources in the lower layer, which is of interest when considering the transition from period 2 to 3 and R2's comment on point 7. Hence, we propose to change the TS diagram along with Chl by the new figure below (attached as Fig. 5).

**R2's point 9:** The speculation using the Hycom model is a useful exercise, but how confident are you in the results? Is the model assimilating the regional float data and regional altimetry?

**AR's point 9:** There is no other way to get a high spatio-temporal resolution picture of the circulation around the island. We are confident in the HYCOM results as the NCODA system (Cummings, 2005; Cummings and Smedstad, 2014) that is used by the HYCOM model, assimilates available satellite altimeter observations, satellite and /in situ/ Sea Surface Temperature (SST) as well as available /in situ/ vertical temperature and salinity profiles (from XBTs, Argo floats and moored buoys). It should be noticed that the Temperature and Salinity profiles from our Argo floats are flagged good or probably good, meaning that these data are likely considered by the reanalysis. However, as the gray list of observations that are not assimilated by the model is not available, we cannot answer definitely to the question.

**R2's point 10:** The authors in the summary again refer to IME effect using data from Tahiti Wake float. Did the authors examine remotely sensed chlorophyll and SSH to provide further characterization of the region? Perhaps there could be some clues.

**AR's response to point 10:** We have examined satellite chlorophyll concentration but without conclusive outcomes. In fact, when examining the monthly and 8-day composites of Chl satellite data from 2002 to 2017 in the studied area, no clear biological enhancement induced by the presence of Tahiti can be evidenced. When considering the IME period observed from the FLeeT float, as explained above (point 4b), it is difficult to have a good interpretation based on satellite data because of the short time scales of the studied process ($\sim 15$ days). For remotely sensed chlorophyll, the best that could be done when focusing on period 2 is shown in the new Figure 1 (see point 4 and Fig. 2 below).

Sea Surface Height have been analyzed with no conclusive outcome about specific dynamics leeward of the island. Moreover, as pointed out in the initial manuscript (lines 29-31 page 9), we also ensured that the FLeeT float did not encountered any eddies that could have induced nutrient uplift and explain the biological enhancement of period 2.

[Figure]

Answers to R2's other comments:

**R2:** Authors have to harmonize the abbreviations in the document
**AR:** We checked and harmonized the abbreviations throughout the document.

**R2:** In Section 3.2, Line 4-5 the following statement is made – "likely reflect differences in phytoplankton community composition (or in the nature of the particle assemblage) in the island wake as compared to the open ocean." Besides $b_{bp}$*, is there any other data like HPLC data from profiles near the Wake float that provide additional insight?
**AR:** Unfortunately, there is no HPLC data near FTWake during the IME period.

**R2:** Figure 2: Units and labels are missing. Isolumes lines are not visible in the CHL plot (panel a) in the PDF submitted for review.
**AR:** We agree that the black dotted lines were barely visible in this figure so we added the PAR time series to Figure 2 (see the new figure attached as Fig. 1 below).

**R2:** Figure 5: Units and label are missing from the figure from c-h. In the legend of the figure 5e and 5f the authors refer that the zoom is in the first 150m but in the plot shows only the first 100m of the water column. In figure 5b the top boxes that indicate the periods are not in same locations as in the others subplots. It will be useful for the reader if the authors can add the MLD in figure 5e and 5f

**AR:** Figure 5 was revised and the figure legend corrected.

**R2:** Are you confident in the nitrate concentrations shown in Figure 5? During late October – early November, you show nitrate concentrations on the order of 0.5-0.8 M after the water column has stratified. And the concentrations seem to be uniform throughout the upper 100 m. These values seem high for an oligotrophic sea, but I have no experience or literature understanding of this region.
**AR:** The limit of detection of NO3- concentration estimated from the SUNA sensor measured from BGC-Argo float is $\sim 1$ micro M (Pasqueron de Fommervault et al., 2015). In our study, we more discuss the dynamics/variation of nitrate concentration than the values itself.

**R2:** Figure 7: The section symbols are small and hard to identify on the map
**AR:** Symbols have been changed.

**R2:** Figure 8: Units and labels are missing
**AR:** Figure 8 was revised.

**R2:** Figure 9: The authors should show the MLD and the DCM. Also, I suggest the authors show all the data from the Tahiti Wake float together.
**AR:** Figure 9 was revised as suggested.
We believe that the common thread of the article is easier to follow keeping Figure 9 separately from Figure 5. Indeed, Figure 5 mostly highlights biogeochemical changes

from period 1 to 2 and 3 over the 150-200 upper meters. It allows us to start the discussion on rain events and island run off as a mechanism to explain changes from period 1 to 2 (Figures 6 and 7). Then, Figure 8 gives insights on the transition from period 2 to 3 through dynamical changes over 0-300 m, which are finally further investigated in Figure 9 and later in the manuscript. Hence, we believe that moving all together the FOpenO figures, followed by all the FLeeT Figures would perturb the common thread of our manuscript. However, according to R1's comment, we propose to add some information at the beginning of this section to clarify our strategy and the plan of the manuscript.

 

Legends of the attached figures below :

**Fig. 1 (new Figure 2 in the revised manuscript):** FOpenO vertical distribution along time of (a) Chl ($mg\ m^{-3}$) with the 0.1, 0.2 and 0.3 $mg\ m^{-3}$ isocontours as dotted black lines, (b) PAR ($mol\ photons\ m^{-2}\ d^{-1}$) with the the 0.1, 1 and 10 $mol\ photons\ m^{-2}\ d^{-1}$ isolumes as dotted black lines, (c) POC ($mg\ m^{-3}$) and (d) density ($kg\ m^{-3}$) with isopycnes (interval = 0.5 $kg\ m^{-3}$, dotted black lines). In each panel, the white and red lines show the MLD and the depth of the DCM, respectively.

**Fig.2 (new Figure 1 in the revised manuscript):** (a) Spatial distribution of OC-CCI surface satellite Chl ($mg\ m^{-3}$) in December 2015 for the Pacific Ocean. Our studied area and the Tahiti island are represented by the white rectangle and the white point respectively. The white star represents the geographic area of the M2014 reference study (see Section 3.1). The trajectories of (b) FOpenO float and (c) FLeeT float are shown with color in background representing the OC-CCI surface satellite Chl ($mg\ m^{-3}$) distribution in December 2015. In panel b FOpenO profiles concomitant

with FLeeT profiles acquisition are colored in red. Grey pixels represent missing data because of clouds or the 500 m bathymetric mask that removed nearshore pixels that could be biased by land. Islands and continents are indicated in black. The Moorea and Tahiti islands are indicated and the mean direction of the South Equatorial Current (SEC) is indicated by the arrow in panel c.

**Fig. 3:** Yearly Vertical distribution of monthly average (from top to bottom): Chl ($mg\ m^{-3}$) and POC ($mg\ m^{-3}$). The left column represents the FOpenO observations and the right column represents the FTLee observations. The color code represents months.

**Fig.4:** Time series of temperature and salinity from FLeeT float for the 0-350 m layer (top panels) and the 0-100 m layer (bottom panels).

**Fig.5: new Figure 6 in the revised manuscript):** T-S diagrams issued from FLeeT during period 2 (a and c panels) and period 3 (b and d panels) as defined in the text (see Section 3.3.1) and Figure 5. Black dotted lines represent isopycnal surfaces (interval= 0.5 $kg\ m^{-3}$). As shown in color, T-S measurements are associated with O2 concentrations ( $mol\ kg^{-3}$, a and b panels) and NO3- concentrations ( $mol\ kg^{-3}$, c and d panels).

a)

b)

c)

d)

**Fig. 1.** New Figure 2 (see legend above)

[Figure]

**Fig. 2.** New Figure 1 (see legend above)

Fig. 3.

**Fig. 4.**

**Fig. 5.** New Figure 6 (see legend above)